# Learning to Augment Distributions for Out-of-Distribution Detection

**Qizhou Wang**[1][*]   **Zhen Fang**[2][*]   **Yonggang Zhang**[1]   **Feng Liu**[3]   **Yixuan Li**[4]   **Bo Han**[1][†]

[1]Department of Computer Science, Hong Kong Baptist University
[2]Australian Artificial Intelligence Institute, University of Technology Sydney
[3]School of Computing and Information Systems, The University of Melbourne
[4]Department of Computer Sciences, University of Wisconsin-Madison
{csqzwang, csygzhang, bhanml}@comp.hkbu.edu.hk
zhen.fang@uts.edu.au    fengliu.ml@gmail.com    sharonli@cs.wisc.edu

## Abstract

Open-world classification systems should discern out-of-distribution (OOD) data whose labels deviate from those of in-distribution (ID) cases, motivating recent studies in OOD detection. Advanced works, despite their promising progress, may still fail in the open world, owing to the lack of knowledge about unseen OOD data in advance. Although one can access auxiliary OOD data (distinct from unseen ones) for model training, it remains to analyze how such auxiliary data will work in the open world. To this end, we delve into such a problem from a learning theory perspective, finding that the distribution discrepancy between the auxiliary and the unseen real OOD data is the key to affecting the open-world detection performance. Accordingly, we propose *Distributional-Augmented OOD Learning* (DAL), alleviating the OOD distribution discrepancy by crafting an *OOD distribution set* that contains all distributions in a Wasserstein ball centered on the auxiliary OOD distribution. We justify that the predictor trained over the worst OOD data in the ball can shrink the OOD distribution discrepancy, thus improving the open-world detection performance given only the auxiliary OOD data. We conduct extensive evaluations across representative OOD detection setups, demonstrating the superiority of our DAL over its advanced counterparts. The code is publicly available at: `https://github.com/tmlr-group/DAL`.

## 1   Introduction

Deep learning in the open world often encounters out-of-distribution (OOD) data of which the label space is disjoint with that of the in-distribution (ID) cases (Hendrycks and Gimpel, 2017; Fang et al., 2022). It leads to the well-known OOD detection problem, where the predictor should make accurate predictions for ID data and detect anomalies from OOD cases (Bulusu et al., 2020; Yang et al., 2021). Nowadays, OOD detection has attracted intensive attention in reliable machine learning due to its integral role in safety-critical applications (Cao et al., 2020; Shen et al., 2021).

OOD detection remains challenging since predictors can make over-confidence predictions for OOD data (Hendrycks et al., 2019), motivating recent studies towards effective OOD detection. Therein, outlier exposure (Hendrycks et al., 2019; Ming et al., 2022) is among the most potent ones, learning from *auxiliary OOD data* to discern ID and OOD patterns. However, due to the openness of the OOD task objective (Wang et al., 2023), auxiliary OOD data can arbitrarily differ from the (unseen) real OOD data in the open world. So, to formally understand their consequences, we model the difference

---

[*]Equal contributions.
[†]Correspondence to Bo Han (bhanml@comp.hkbu.edu.hk).

37th Conference on Neural Information Processing Systems (NeurIPS 2023).

between auxiliary and real OOD data by their distribution discrepancy, measured by the Wasserstein distance (Villani, 2021, 2008). Then, we reveal the negative impacts of such OOD distribution discrepancy on the real detection power, with a larger distribution discrepancy indicating a lower performance on real OOD data, cf., Eq. (4).

The OOD distribution discrepancy threatens the open-world detection performance for outlier exposure. Therefore, we raise a natural question in *how to alleviate such an OOD distribution discrepancy*. Hence, this paper establishes a promising learning framework named *Distributional-Augmented OOD Learning* (DAL). Therein, we augment the auxiliary OOD distribution by crafting an *OOD*

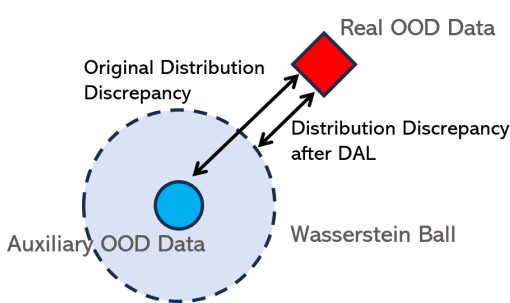

Figure 1: A heuristic illustration for our DAL. A large distribution discrepancy between the auxiliary and the unseen OOD data will hurt the real detection effectiveness. However, by ensuring uniformly well performance inside the Wasserstein ball, we can mitigate the distribution discrepancy and thus improve the detection power in the open world.

*distribution set* containing all distributions in a Wasserstein ball (Villani, 2021, 2008), centered on the auxiliary OOD distribution. Then, by making the predictor learn from the worst OOD distribution in the set, cf., Eq. (8), one can alleviate the negative impacts of the distribution discrepancy. Moreover, our proposed framework enjoys the learning guarantees towards the expected risk with respect to the real OOD distribution, making OOD detection stay effective when facing unseen data (cf., Theorem 3). Figure 1 provides a conceptual explanation: learning from the worst OOD distribution ensures the uniformly well performance inside the Wasserstein ball, enlarging the influence of the auxiliary OOD distribution. Thus, one can shrink the OOD distribution discrepancy between the auxiliary and the real OOD data and improve OOD detection.

In realization, the primal learning objective in Eq. (8) is generally intractable due to the infinite-dimensional optimization for the worst OOD distribution search. Instead, we adopt the dual form with respect to the original learning problem (cf., Theorem 1), transforming it into a tractable problem of the worst OOD data search in a finite-dimensional space. Furthermore, following Du et al. (2022a); Mehra et al. (2022), the data search procedure is conducted in the embedding space, which can benefit the open-world performance of OOD detection with decent costs of additional computation.

We conduct extensive experiments over representative OOD detection setups, revealing the open-world performance of our method toward effective OOD detection. For example, our DAL reduces the average FPR95 by 1.99 to 13.46 on CIFAR benchmarks compared with the conventional outlier exposure (Hendrycks et al., 2019). Overall, we summarize our contributions into three folds:

- We measure the difference between the auxiliary and the real OOD data by the Wasserstein distance, and establish an effective learning framework, named DAL, to mitigate the OOD distribution discrepancy issue. We further guarantee our performance with respect to unseen real OOD data via Theorem 3, which is new to previous works.

- DAL leads to a practical method in Algorithm 1, learning from the worst cases in the Wasserstein ball to improve the open-world detection performance. Overall, our method solves the dual problem, which performs the worst-case search in the embedding space, which is simple to compute yet effective in OOD detection.

- We conduct extensive experiments in Section 5 to evaluate our effectiveness, ranging from the well-known CIFAR benchmarks to the challenging ImageNet settings. The empirical results comprehensively demonstrate our superiority over advanced counterparts, and the improvement is mainly attributed to our distributional-augmented learning framework.

A detailed overview of existing OOD detection methods and theories can be found in Appendix A, and a summary of the important notations can be found in Appendix B.

## 2 Outlier Exposure

Let $\mathcal{X}$ denote the feature space and $\mathcal{Y} = \{1, \ldots, C\}$ denote the label space with respect to the ID distribution. We consider the ID distribution $D_{X_I Y_I}$, a joint distribution defined over $\mathcal{X} \times \mathcal{Y}$, where $X_I$ and $Y_I$ are random variables whose outputs are from spaces $\mathcal{X}$ and $\mathcal{Y}$. We also have an OOD joint distribution $D_{X_O Y_O}$, where $X_O$ is a random variable from $\mathcal{X}$, but $Y_O$ is a random variable whose outputs do not belong to $\mathcal{Y}$, i.e., $Y_O \notin \mathcal{Y}$ (Fang et al., 2022).

The classical OOD detection (Hendrycks and Gimpel, 2017; Yang et al., 2021) typically considers an open-world setting, where the real OOD data drawn from $D_{X_O Y_O}$ are unseen during training. Recently, Fang et al. (2022) have provided several *strong* conditions necessary to ensure the success of the classical OOD setting. Furthermore, to increase the possibility of success for OOD detection and weaken the strong conditions proposed by Fang et al. (2022), advanced works (Hendrycks et al., 2019; Chen et al., 2021) introduce a promising approach named *outlier exposure*, where a set of auxiliary OOD data is employed as a surrogate of real OOD data. Here, we provide a formal definition.

**Problem 1** (OOD Detection with Outlier Exposure). Let $D_{X_I Y_I}$, $D_{X_O}$, and $D_{X_A}$ be the ID joint distribution, the OOD distribution, and the auxiliary OOD distribution, respectively. Given the sets of samples called the ID and the auxiliary OOD data, namely,

$$S = \{(\mathbf{x}_I^1, y_I^1), ..., (\mathbf{x}_I^n, y_I^n)\} \sim D_{X_I Y_I}^n, \ i.i.d., \quad T = \{\mathbf{x}_A^1, ..., \mathbf{x}_A^m\} \sim D_{X_A}^m, \ i.i.d.,$$

outlier exposure trains a predictor $\mathbf{f}$ by using the training data $S$ and $T$, such that for any test data $\mathbf{x}$: 1) if $\mathbf{x}$ is an observation from $D_{X_I}$, the predictor $\mathbf{f}$ can classify $\mathbf{x}$ into its correct ID label; otherwise 2) if $\mathbf{x}$ is an observation from $D_{X_O}$, the predictor $\mathbf{f}$ can detect $\mathbf{x}$ as an OOD case.

**OOD Scoring.** Many existing methods detect OOD data by using various score-based strategies (Hendrycks and Gimpel, 2017; Lee et al., 2018a; Liu et al., 2020; Sun et al., 2022). In general, given a model $\mathbf{f} : \mathcal{X} \to \mathbb{R}^C$ and a scoring function $s(\cdot; \mathbf{f}) : \mathcal{X} \to \mathbb{R}$, the OOD detector $g_\lambda$ is given by:

$$g_\lambda(\mathbf{x}) = \text{ID}, \ \text{if } s(\mathbf{x}; \mathbf{f}) \geq \lambda; \text{ otherwise, } g_\lambda(\mathbf{x}) = \text{OOD},$$

where $\lambda$ is a given threshold. For example, as a well-known baseline scoring function, the maximum softmax prediction (MSP) (Hendrycks and Gimpel, 2017) is given by:

$$s_{\text{MSP}}(\mathbf{x}; \mathbf{f}) = \max_{k \in \mathcal{Y}} \texttt{softmax}_k \, \mathbf{f}(\mathbf{x}), \tag{1}$$

with $\texttt{softmax}_k(\cdot)$ denoting the $k$-th dimension of the softmax output.

**Model and Risks.** We denote $\mathbf{f_w} : \mathcal{X} \to \mathbb{R}^C$ the predictor with parameters $\mathbf{w} \in \mathcal{W}$, with $\mathcal{W}$ the parameter space. We consider the loss functions $\ell$ and $\ell_{\text{OE}}$ w.r.t. the ID and the OOD cases, respectively. Then, the expected and the empirical *ID risks* of the model $\mathbf{f_w}$ can be written as:

$$R_I(\mathbf{w}) = \mathbb{E}_{(\mathbf{x},y) \sim D_{X_I Y_I}} \ell(\mathbf{f_w}; \mathbf{x}, y) \ \text{ and } \ \widehat{R}_I(\mathbf{w}) = \frac{1}{n} \sum_{i=1}^{n} \ell(\mathbf{f_w}; \mathbf{x}_I^i, y_I^i).$$

The expected and the empirical *auxiliary OOD risks* are then given by

$$R_A(\mathbf{w}) = \mathbb{E}_{\mathbf{x} \sim D_{X_A}} \ell_{\text{OE}}(\mathbf{f_w}; \mathbf{x}) \ \text{ and } \ \widehat{R}_A(\mathbf{w}) = \frac{1}{m} \sum_{i=1}^{m} \ell_{\text{OE}}(\mathbf{f_w}; \mathbf{x}_A^i),$$

and the expected *real OOD risk* is given by $R_O(\mathbf{w}) = \mathbb{E}_{\mathbf{x} \sim D_{X_O}} \ell_{\text{OE}}(\mathbf{f_w}; \mathbf{x})$. Accordingly, we can define the expected *detection risk* with respect to real OOD data, following

$$R_D(\mathbf{w}) = R_I(\mathbf{w}) + \alpha R_O(\mathbf{w}), \tag{2}$$

where $\alpha$ is the trade-off parameter.

**Learning Strategy.** After the scoring function is selected, one can obtain the OOD detector if the model $\mathbf{f_w}$ is given. Under the Problem 1 of outlier exposure, a common learning strategy is to optimize the empirical ID and auxiliary OOD risk jointly (Hendrycks et al., 2019), namely,

$$\min_{\mathbf{w} \in \mathcal{W}} \left[ \widehat{R}_I(\mathbf{w}) + \alpha \widehat{R}_A(\mathbf{w}) \right]. \tag{3}$$

Note that the auxiliary OOD data are employed in Eq. (3), which can arbitrarily differ from the real OOD cases. Then, it is generally expected that the predictor $\mathbf{f_w}$, trained over the auxiliary OOD data, can perform well even on unseen OOD data, i.e., a small value of $R_D(\mathbf{w})$ is expected.

## 3 Motivation

To the general learning strategy in Eq. (3), intuitively, if the auxiliary data are sampled from a distribution similar to real ones, the predictor will perform well for real OOD data. However, auxiliary and real OOD data differ in practice, posing us to suspect their open-world detection performance. To formally study the problem, we measure the difference between auxiliary and real OOD data in the distribution level, motivating our discussion of *OOD distribution discrepancy*.

**Distribution Discrepancy.** In this paper, we adopt a classical measurement for the distribution discrepancy—Optimal Transport Cost (Sinha et al., 2018; Mehra et al., 2022).

**Definition 1** (Optimal Transport Cost and Wasserstein-1 Distance (Villani, 2021, 2008)). *Given a cost function $c : \mathcal{X} \times \mathcal{X} \to \mathbb{R}_+$, the optimal transport cost between two distributions $D$ and $D'$ is*

$$W_c(D, D') = \inf_{\pi \in \Pi(D, D')} \mathbb{E}_{(\mathbf{x}, \mathbf{x}') \sim \pi} c(\mathbf{x}, \mathbf{x}'),$$

*where $\Pi(D, D')$ is the space of all couplings for $D$ and $D'$. Furthermore, if the cost $c$ is a metric, then the optimal transport cost is also called the Wasserstein-1 distance.*

Based on Definition 1, we use the distribution discrepancy to measure the difference between the auxiliary and the real OOD data, namely, $W_c(D_{X_O}, D_{X_A})$. Then, we can formally study the impacts of such a discrepancy on the detection performance of the predictor. Under certain assumptions (cf., Corollary 1), we can prove that with high probability, the following generalization bound holds:

$$R_D(\widehat{\mathbf{w}}) \leq \min_{\mathbf{w} \in \mathcal{W}} \left( R_I(\mathbf{w}) + \alpha R_A(\mathbf{w}) \right) + \alpha L_c W_c(D_{X_O}, D_{X_A}) + \mathcal{O}(1/\sqrt{n}) + \mathcal{O}(1/\sqrt{m}), \quad (4)$$

where $\widehat{\mathbf{w}}$ is the parameter learned by Eq. (3), i.e., $\widehat{\mathbf{w}} \in \arg\min_{\mathbf{w} \in \mathcal{W}} \widehat{R}_I(\mathbf{w}) + \alpha \widehat{R}_A(\mathbf{w})$, $L_c$ is the Lipschitz constant of $\ell_{OE}$ w.r.t. the cost function $c(\cdot, \cdot)$ (see Theorem 3). In general, the expected detection risk $R_D(\widehat{\mathbf{w}})$ measures the expected performance on unseen OOD data given the predictor trained on the auxiliary OOD data. Then, due to the upper bound, the impacts of the OOD distribution discrepancy are reflected by the Wasserstein-1 distance between the auxiliary and the real OOD data, i.e., $W_c(D_{X_O}, D_{X_A})$. Therefore, although classical outlier exposure can improve OOD detection to some extent, it fails to ensure reliable detection of unseen OOD data, in that a larger distribution discrepancy generally indicates a worse guarantee for open-world OOD detection.

The key to improve the detection performance is mitigating the negative impact induced by the OOD distribution discrepancy. To tackle this problem, a simple lemma inspires us:

**Lemma 1.** *Let $d(\cdot, \cdot)$ be the distance to measure the discrepancy between distributions. Given a space $\mathfrak{D}$ consisting of some OOD distributions, if $D_{X_A} \in \mathfrak{D}$, then*

$$\inf_{D_{X'} \in \mathfrak{D}} d(D_{X'}, D_{X_O}) \leq d(D_{X_A}, D_{X_O}). \quad (5)$$

*If $d(\cdot, \cdot)$ is the Optimal Transport Cost in Definition 1, the cost function $c$ is a continuous metric, and $\mathfrak{D}$ is the Wasserstein-1 ball with a radius $\rho > 0$, i.e., $\mathfrak{D} = \{D_{X'} : W_c(D_{X'}, D_{X_A}) \leq \rho\}$, then*

$$\inf_{D_{X'} \in \mathfrak{D}} W_c(D_{X'}, D_{X_O}) \leq \max\{W_c(D_{X_A}, D_{X_O}) - \rho, 0\}. \quad (6)$$

In the light of Lemma 1, we introduce a specific set of distributions $\mathfrak{D}$, augmented around the auxiliary OOD distribution. It makes it possible to mitigate the distribution discrepancy, following Eqs. (5) and (6). Therefore, instead of choosing a model $\mathbf{f_w}$ that directly minimizes the empirical risk in Eq. (3), we target augmenting the auxiliary OOD data within the distribution space $\mathfrak{D}$, namely,

$$\min_{\mathbf{w} \in \mathcal{W}} \left[ \widehat{R}_I(\mathbf{w}) + \alpha \sup_{D_{X'} \in \mathfrak{D}} \mathbb{E}_{\mathbf{x} \sim D_{X'}} \ell_{OE}(\mathbf{f_w}; \mathbf{x}) \right], \text{ subject to } \widehat{D}_{X_A} \in \mathfrak{D}, \quad (7)$$

where $\widehat{D}_{X_A}$ is the empirical form of $D_{X_A}$, i.e., $\widehat{D}_{X_A} = \frac{1}{m} \sum_{i=1}^m \delta_{\mathbf{x}_A^i}$ and $\delta_{\mathbf{x}_A^i}$ is the dirac measure.

## 4 Learning Framework

This section proposes a general learning framework to mitigate the OOD distribution discrepancy. As aforementioned, we consider an augmented set of OOD distributions to improve OOD detection, thus named *Distributional-Augmented OOD Learning* (DAL).

To begin with, we need to select a suitable distribution space $\mathfrak{D}$ for the tractable solutions of Eq. (7). Generally, the choice of $\mathfrak{D}$ influences both the richness of the auxiliary data as well as the tractability of the resulting optimization problem. Previous works have developed a series of distribution spaces, e.g., the distribution ball based on $f$-divergences (Namkoong and Duchi, 2016; Michel et al., 2021) and maximum mean discrepancy (MMD) (Staib and Jegelka, 2019). However, there are several drawbacks for the distribution balls based on $f$-divergences and MMD: 1) any $f$-divergence-based space $\mathfrak{D}$ contains only distributions within the same support set as $\widehat{D}_{X_A}$; and 2) the effective solutions in the MMD-based space have not been provided (Staib and Jegelka, 2019).

Instead, motivated by Sinha et al. (2018); Mehra et al. (2022); Dai et al. (2023) and Theorem 1, we consider the Wasserstein ball. For any $\rho > 0$, we define the augmented OOD distribution set as

$$\mathfrak{D} = \{D_{X'} : W_c(D_{X'}, \widehat{D}_{X_A}) \le \rho\},$$

and consider the following optimization problem:

$$\min_{\mathbf{w} \in \mathcal{W}} \widehat{R}_D(\mathbf{w}; \rho) = \min_{\mathbf{w} \in \mathcal{W}} \left[ \widehat{R}_I(\mathbf{w}) + \alpha \widehat{R}_O(\mathbf{w}; \rho) \right], \tag{8}$$

where

$$\widehat{R}_O(\mathbf{w}; \rho) = \sup_{W_c(D_{X'}, \widehat{D}_{X_A}) \le \rho} \mathbb{E}_{\mathbf{x} \sim D_{X'}} \ell_{OE}(\mathbf{f}_\mathbf{w}; \mathbf{x}). \tag{9}$$

However, the optimization problem in Eq. (8) is intractable due to the infinite-dimensional search for the distribution $D_{X'}$. Fortunately, the following dual theorem provides a solution:

**Theorem 1** (Blanchet and KarthyekRajhaaA. (2016)). *Let $c(\cdot, \cdot)$ be a continuous metric and $\phi_\gamma(\mathbf{w}; \mathbf{x}) = \sup_{\mathbf{x}' \in \mathcal{X}} \{\ell(\mathbf{f}_\mathbf{w}; \mathbf{x}') - \gamma c(\mathbf{x}', \mathbf{x})\}$ be the robust surrogate function. Then, for any $\rho > 0$,*

$$\widehat{R}_D(\mathbf{w}; \rho) = \widehat{R}_I(\mathbf{w}) + \alpha \inf_{\gamma \ge 0} \left\{ \gamma \rho + \frac{1}{m} \sum_{i=1}^m \phi_\gamma(\mathbf{w}; \mathbf{x}_A^i) \right\}. \tag{10}$$

Theorem 1 provides a feasible surrogate for the original optimization problem in Eq. (8), transforming the infinite-dimensional problem to its finite counterpart, i.e., the data feature search. We use Eq. (10) to design our algorithm, cf., Section 4.2.

### 4.1 Theoretical Supports

This section provides the theoretical support for our DAL. Specifically, 1) Theorem 2 shows that the empirical model given by Eq. (8) can achieve consistent learning performance, and 2) Theorem 3 further demonstrates the expected detection risk estimation, i.e., $R_D(\mathbf{w})$, with respect to the empirical model given by Eq. (8). All the proofs can be found in Appendix C. To state our theoretical results, we use the notation $R_D(\mathbf{w}; \rho)$ to represent the ideal form of $\widehat{R}_D(\mathbf{w}; \rho)$, which is defined by

$$R_D(\mathbf{w}; \rho) = R_I(\mathbf{w}) + \alpha R_O(\mathbf{w}; \rho),$$

where

$$R_O(\mathbf{w}; \rho) = \sup_{W_c(D_{X'}, D_{X_A}) \le \rho} \mathbb{E}_{\mathbf{x} \sim D_{X'}} \ell_{OE}(\mathbf{f}_\mathbf{w}; \mathbf{x}).$$

Similar to Sinha et al. (2018), our results rely on the covering number (cf., Appendix C.1) for the model classes $\mathcal{F} = \{\ell(\mathbf{f}_\mathbf{w}; \cdot) : \mathbf{w} \in \mathcal{W}\}$ and $\mathcal{F}_{OE} = \{\ell_{OE}(\mathbf{f}_\mathbf{w}; \cdot) : \mathbf{w} \in \mathcal{W}\}$ to represent their complexity. Intuitively, the covering numbers $\mathcal{N}(\mathcal{F}, \epsilon, L^\infty)$ and $\mathcal{N}(\mathcal{F}_{OE}, \epsilon, L^\infty)$ are the minimal numbers of $L^\infty$ balls of radius $\epsilon > 0$ needed to cover the model classes $\mathcal{F}$ and $\mathcal{F}_{OE}$, respectively. Now, we demonstrate that DAL can achieve consistent performance under mild assumptions.

**Theorem 2** (Excess Generalization Bound). *Assume that $0 \le \ell(\mathbf{f}_\mathbf{w}; \mathbf{x}, y) \le M_\ell$, $0 \le \ell_{OE}(\mathbf{f}_\mathbf{w}; \mathbf{x}) \le M_{\ell_{OE}}$, and $c(\cdot, \cdot) : \mathcal{X} \times \mathcal{X} \to \mathbb{R}_+$ is a continuous metric. Let $\widehat{\mathbf{w}}$ be the optimal solution of Eq. (8), i.e., $\widehat{\mathbf{w}} \in \arg\min_{\mathbf{w} \in \mathcal{W}} \widehat{R}_D(\mathbf{w}; \rho)$. Then with the probability at least $1 - 4e^{-t} > 0$,*

$$R_D(\widehat{\mathbf{w}}; \rho) - \min_{\mathbf{w} \in \mathcal{W}} R_D(\mathbf{w}; \rho) \le \epsilon(n, m; t), \tag{11}$$

*for any $\rho > 0$, where*

$$\epsilon(n, m; t) = \frac{b_0 M_\ell}{\sqrt{n}} \int_0^1 \sqrt{\log \mathcal{N}(\mathcal{F}, M_\ell \epsilon, L^\infty)} d\epsilon + 2M_\ell \sqrt{\frac{2t}{n}}$$

$$+ \alpha b_1 \sqrt{\frac{M_{\ell_{OE}}^3}{\rho^2 m}} \int_0^1 \sqrt{\log \mathcal{N}(\mathcal{F}_{OE}, M_{\ell_{OE}} \epsilon, L^\infty)} d\epsilon + \alpha b_2 M_{\ell_{OE}} \sqrt{\frac{2t}{m}},$$

*where $b_0$, $b_1$ and $b_2$ are uniform constants.*

---

**Algorithm 1** Distributional-Augmented OOD Learning (DAL)

---
**Input:** ID and OOD samples from $D_{X_{\mathrm{I}}Y_{\mathrm{I}}}$ and $D_{X_{\mathrm{A}}}$;
  **for** st $= 1$ **to** num_step **do**
    Sample $S_{\mathrm{B}}$ and $T_{\mathrm{B}}$ from $D_{X_{\mathrm{I}}Y_{\mathrm{I}}}$ and $D_{X_{\mathrm{A}}}$;
    Initialize $\mathbf{p}^i \sim \mathcal{N}(\mathbf{0}, \sigma I)$, $\forall i \in \{1, \ldots, |T_{\mathrm{B}}|\}$;
    **for** se $= 1$ **to** num_search **do**
      $\boldsymbol{\psi}^i = \nabla_{\mathbf{p}^i} \left[ \ell_{\mathrm{OE}} \left( \mathbf{h}(\mathbf{e}(\mathbf{x}_{\mathrm{A}}^i + \mathbf{p}^i); \mathbf{e}(\mathbf{x}_{\mathrm{A}}^i)) - \gamma \left\| \mathbf{p}^i \right\|_1 \right] \right.$, $\forall i \in \{1, \ldots, |T_{\mathrm{B}}|\}$;
      $\mathbf{p}^i \leftarrow \mathbf{p}^i + \mathrm{ps}\boldsymbol{\psi}^i$, $\forall i \in \{1, \ldots, |T_{\mathrm{B}}|\}$
    **end for**
    $\gamma \leftarrow \min \left( \max \left( \gamma - \beta(\rho - \frac{1}{|T_{\mathrm{B}}|} \sum_{i=1}^{|T_{\mathrm{B}}|} \|\mathbf{p}^i\|, \gamma_{\max}), 0 \right) \right.$;
    $\mathbf{w} \leftarrow \mathbf{w} - \mathrm{lr}\nabla_{\mathbf{w}} \left[ \frac{1}{|T_{\mathrm{B}}|} \sum_{i=1}^{|T_{\mathrm{B}}|} \ell_{\mathrm{OE}}(\mathbf{h}(\mathbf{g}(\mathbf{x}_{\mathrm{A}}^i) + \mathbf{p}^i)) + \alpha \frac{1}{|S_{\mathrm{B}}|} \sum_{i=1}^{|S_{\mathrm{B}}|} \ell(\mathbf{f}_{\mathbf{w}}; \mathbf{x}_{\mathrm{I}}^i, y_{\mathrm{I}}^i)) \right]$;
  **end for**
**Output:** model parameter $\mathbf{w}$.

---

Furthermore, under proper conditions, one can show that the bound in Eq. (11) can attain $\mathcal{O}(1/\sqrt{n}) + \mathcal{O}(1/\sqrt{m})$, i.e., $R_D(\widehat{\mathbf{w}}; \rho) - \min_{\mathbf{w} \in \mathcal{W}} R_D(\mathbf{w}; \rho) \leq \mathcal{O}(1/\sqrt{n}) + \mathcal{O}(1/\sqrt{m})$. Corollary 1 in Appendix C.5 gives an example to support the above claim. Next, we give a learning bound to estimate the expected detection risk in Eq. (2) w.r.t. the model $\mathbf{f}_{\widehat{\mathbf{w}}}$ given by Eq. (8).

**Theorem 3** (Risk Estimation). *Given the same conditions in Theorem 2 and let $\widehat{\mathbf{w}}$ be the solution of Eq. (8), which is given by $\widehat{\mathbf{w}} \in \arg\min_{\mathbf{w} \in \mathcal{W}} \widehat{R}_D(\mathbf{w}; \rho)$. If $\ell_{\mathrm{OE}}(\mathbf{f}_{\mathbf{w}}; \mathbf{x})$ is $L_c$-Lipschitz w.r.t. $c(\cdot, \cdot)$, i.e., $|\ell_{\mathrm{OE}}(\mathbf{f}_{\mathbf{w}}; \mathbf{x}) - \ell_{\mathrm{OE}}(\mathbf{f}_{\mathbf{w}}; \mathbf{x}')| \leq L_c c(\mathbf{x}, \mathbf{x}')$, then with the probability at least $1 - 4e^{-t} > 0$,*

$$R_D(\widehat{\mathbf{w}}) - \overbrace{\min_{\mathbf{w} \in \mathcal{W}} R_D(\mathbf{w}; \rho)}^{\text{approximate risk}} \leq \underbrace{\alpha L_c \max\{\mathrm{W}_c(D_{X_{\mathrm{O}}}, D_{X_{\mathrm{A}}}) - \rho, 0\} + \epsilon(n, m; t)}_{\text{estimation error}},$$

*for any $\rho > 0$, where $\epsilon(n, m; t)$ is defined in Theorem 2.*

The bias term $\alpha L_c \max\{\mathrm{W}_c(D_{X_{\mathrm{O}}}, D_{X_{\mathrm{A}}}) - \rho, 0\} = 0$ when $\rho$ is large enough. Hence, a large $\rho$ implies a small estimation error. Although a larger $\rho$ leads to better generalization ability, the approximate risk $\min_{\mathbf{w} \in \mathcal{W}} R_D(\mathbf{w}; \rho)$ may become larger. It implies that for practical effectiveness, i.e., small $R_D(\widehat{\mathbf{w}})$, there is a trade-off between the approximate risk $\min_{\mathbf{w} \in \mathcal{W}} R_D(\mathbf{w}; \rho)$ and the bias $\alpha L_c \max\{\mathrm{W}_c(D_{X_{\mathrm{O}}}, D_{X_{\mathrm{A}}}) - \rho, 0\}$ across different choices of $\rho$. Hence, we need to choose a proper $\rho$ for open-world detection with unseen data (cf., Section 5.3).

### 4.2 Proposed Algorithm

In this section, we introduce the algorithm design for DAL, summarized in Algorithm 1. Due to the space limit, we provide further discussions in Appendix E.

**Losses and Cost Function.** Following Hendrycks et al. (2019), we adopt the cross entropy loss to realize $\ell$ and the KL-divergence between model predictions and uniform distribution for $\ell_{\mathrm{OE}}$. We also define the cost function $c$ by the $l_1$ norm, namely, $c(\mathbf{x}, \mathbf{x}') = \|\mathbf{x} - \mathbf{x}'\|_1$.

**Algorithm Design.** By Theorem 1, we can address the primary problem in Eq. (8) by the dual problem in Eq. (9). Additionally, following Du et al. (2022a), we perturb for the worst OOD data in the embedding space. Denote the model $\mathbf{f}_{\mathbf{w}} = \mathbf{h} \circ \mathbf{e}$ with $\mathbf{h}$ the classifier and $\mathbf{e}$ the feature extractor, we find the perturbation $\mathbf{p}$ for the embedding features, i.e., $\mathbf{e}(\mathbf{x})$, of the associated data $\mathbf{x}$. The perturbation $\mathbf{p}$ should lead to the worst OOD case for the surrogate function in Theorem 1, namely,

$$\phi_{\gamma}(\mathbf{w}; \mathbf{e}(\mathbf{x})) = \sup_{\mathbf{p} \in \mathcal{E}} \{\ell_{\mathrm{OE}}(\mathbf{h}(\mathbf{e}(\mathbf{x}) + \mathbf{p}); \mathbf{e}(\mathbf{x})) - \gamma\|\mathbf{p}\|_1\},$$

where $\mathcal{E}$ denotes the space of embedding features. Note that we abuse the definition of $\ell_{\mathrm{OE}}$, emphasizing that we perturb the embedding features of $\mathbf{e}(\mathbf{x})$ by $\mathbf{p}$.

**Training and Inference.** Our definition of $\phi_{\gamma}(\mathbf{w}; \mathbf{e}(\mathbf{x}))$ leads to a particular realization of Eq. (10), which is the learning objective of our DAL. It can be solved by stochastic gradient optimization for deep models, e.g., mini-batch stochastic gradient descent. After training, we use the MSP scoring function by default and discuss the possibility of other scoring functions in Appendix F.3.

**Stochastic Realization.** Algorithm 1 gives a stochastic realization of DAL, where ID and auxiliary OOD mini-batches are randomly sampled in each stochastic iteration, denoted by $S_B$ and $T_B$ respectively. Therein, we first find the perturbation $\mathbf{p}$ that leads to the maximal $\phi_\gamma(\mathbf{w}, \mathbf{e}(\mathbf{x}))$. The value of $\mathbf{p}$ is initialized by random Gaussian noise with the standard deviation $\sigma$ and updated by gradient ascent for `num_search` steps with the perturbation strength `ps`. Then we update $\gamma$ by one step of gradient descent with the learning rate $\beta$, and further clipping between 0 and $\gamma_{max}$ to avoid extreme values. Finally, given the proper perturbations for the auxiliary OOD data in $T_B$, we update the model parameter $\mathbf{w}$ by one step of mini-batch gradient descent.

# 5 Experiments

In this section, we mainly test DAL on the CIFAR (Krizhevsky and Hinton, 2009) benchmarks (as ID datasets). To begin with, we introduce the evaluation setups.

**OOD Datasets.** We adopt the 80 Million Tiny Images (Torralba et al., 2008) as the auxiliary OOD dataset; Textures (Cimpoi et al., 2014), SVHN (Netzer et al., 2011), Places365 (Zhou et al., 2018), LSUN (Yu et al., 2015), and iSUN (Xu et al., 2015) as the (test-time) real OOD datasets. We eliminate those data whose labels coincide with ID cases.

**Pre-training Setups.** We employ Wide ResNet-40-2 (Zagoruyko and Komodakis, 2016) trained for 200 epochs via empirical risk minimization, with a batch size 64, momentum 0.9, and initial learning rate 0.1. The learning rate is divided by 10 after 100 and 150 epochs.

**Hyper-parameters Tuning Strategy.** The hyper-parameters are tuned based on the validation data, separated from the training ID and auxiliary OOD data, which is a common strategy in OOD detection with outlier exposure field (Hendrycks et al., 2019; Chen et al., 2021). Specifically, we fix $\sigma = 0.001$, `num_search` $= 10$, and adopt the grid search to choose $\gamma_{max}$ from $\{0.1, 0.5, 1, 5, 10, 50\}$; $\beta$ from $\{1e^{-3}, 5e^{-3}, 1e^{-2}, 5e^{-2}, 1e^{-1}, 5e^{-1}, 1, 5\}$; $\rho$ from $\{1e^{-2}, 1e^{-1}, 1, 10, 100\}$; `ps` from $\{1e^{-3}, 1e^{-2}, 1e^{-1}, 1, 10, 100\}$; $\alpha$ from $\{0.1, 0.5, 1.0, 1.5, 2.0\}$.

**Hyper-parameters Setups.** For CIFAR-10, DAL is run for 50 epochs with the ID batch size 128, the OOD batch size 256, the initial learning rate 0.07, $\gamma_{max} = 10$, $\beta = 0.01$, $\rho = 10$, `ps` $= 1$, and $\alpha = 1$. For CIFAR-100, DAL is run for 50 epochs with the ID batch size 128, the OOD batch size 256, the initial learning rate 0.07, $\gamma_{max} = 10$, $\beta = 0.005$, $\rho = 10$, and `ps` $= 1$, and $\alpha = 1$. For both cases, we employ cosine decay (Loshchilov and Hutter, 2017) for the model learning rate.

**Baseline Methods.** We compare DAL with representative methods, including MSP (Hendrycks and Gimpel, 2017), Free Energy (Liu et al., 2020), ASH (Djurisic et al., 2023), ReAct (Sun et al., 2021), Mahalanobis (Lee et al., 2018a), KNN (Sun et al., 2022), KNN+ (Sun et al., 2022), CSI (Tack et al., 2020), VOS (Du et al., 2022a), Outlier Exposure (OE) (Hendrycks et al., 2019), Energy-OE (Liu et al., 2020), ATOM (Chen et al., 2021), DOE (Wang et al., 2023), and POEM (Ming et al., 2022). We adopt their suggested setups but unify the backbones for fairness.

**Evaluation Metrics.** The detection performance is evaluated via two representative metrics, which are both threshold-independent: the false positive rate of OOD data when the true positive rate of ID data is at 95% (FPR95); and the area under the receiver operating characteristic curve (AUROC), which can be viewed as the probability of the ID case having greater score than that of the OOD case.

Due to the space limit, we test our DAL with more advanced scoring strategies in Appendix F.3 and conduct experiments on the more complex ImageNet (Deng et al., 2009) dataset in Appendix F.10.

## 5.1 Main Results

The main results are summarized in Table 1, where we report the detailed results across the considered real OOD datasets. First, we reveal that using auxiliary OOD data can generally lead to better results than using only ID information, indicating that outlier exposure remains a promising direction worth studying. However, as demonstrated in Section 3, the OOD distribution discrepancy can hurt its open-world detection power, while previous works typically oversee such an important issue. Therefore, our DAL, which can alleviate the OOD distribution discrepancy, reveals a large improvement over the original outlier exposure. Specifically, comparing with the conventional outlier exposure, our method reveals 1.99 and 0.13 average improvements w.r.t. FPR95 and AUROC on the CIFAR-10 dataset, and 13.46 and 3.65 of the average improvements on CIFAR-100 dataset. For advanced works that consider the OOD sampling strategies, e.g., ATOM and POEM, DAL can achieve much better results, especially for the CIFAR-100 case. The reason is that these methods mainly consider the situations

Table 1: Comparison between our method and advanced methods on the CIFAR benchmarks. ↓ (or ↑) indicates smaller (or larger) values are preferred, and a bold font indicates the best result in a column. Methods are grouped based on 1) using ID data only and 2) using additional information about auxiliary OOD data. Two groups are separated by the horizontal line for each ID case.

| Method | SVHN | | LSUN | | iSUN | | Textures | | Places365 | | Average | |
|---|---|---|---|---|---|---|---|---|---|---|---|---|
| | FPR95↓ | AUROC↑ | FPR95↓ | AUROC↑ | FPR95↓ | AUROC↑ | FPR95↓ | AUROC↑ | FPR95↓ | AUROC↑ | FPR95↓ | AUROC↑ |
| CIFAR-10 | | | | | | | | | | | | |
| Using ID data only | | | | | | | | | | | | |
| MSP | 48.89 | 91.97 | 25.53 | 96.49 | 56.44 | 89.86 | 59.68 | 88.42 | 60.19 | 88.36 | 50.15 | 91.02 |
| Free Energy | 35.21 | 91.24 | 4.42 | 99.06 | 33.84 | 92.56 | 52.46 | 85.35 | 40.11 | 90.02 | 33.21 | 91.64 |
| ASH | 33.98 | 91.79 | 4.76 | 98.98 | 34.38 | 92.64 | 50.90 | 86.07 | 40.89 | 89.79 | 32.98 | 91.85 |
| Mahalanobis | 12.21 | 97.70 | 57.25 | 89.58 | 79.74 | 77.87 | 15.20 | 95.40 | 68.81 | 82.39 | 46.64 | 88.59 |
| KNN | 26.56 | 95.93 | 27.52 | 95.43 | 33.55 | 93.15 | 37.62 | 93.07 | 41.67 | 91.21 | 33.38 | 93.76 |
| KNN+ | 3.28 | 99.33 | 2.24 | 98.90 | 17.85 | 95.65 | 10.87 | 97.72 | 30.63 | 94.98 | 12.97 | 97.32 |
| CSI | 17.37 | 97.69 | 6.75 | 98.46 | 12.58 | 97.95 | 25.65 | 94.70 | 40.00 | 92.05 | 20.47 | 96.17 |
| VOS | 36.55 | 93.30 | 9.98 | 98.03 | 28.93 | 94.25 | 52.83 | 85.74 | 39.56 | 89.71 | 33.57 | 92.21 |
| Using ID data and auxiliary OOD data | | | | | | | | | | | | |
| OE | 2.36 | 99.27 | 1.15 | **99.68** | 2.48 | 99.34 | 5.35 | 98.88 | 11.99 | 97.23 | 4.67 | 98.88 |
| Energy-OE | 0.97 | 99.54 | 1.00 | 99.15 | 2.32 | 99.27 | 3.42 | **99.18** | 9.57 | 97.44 | 3.46 | 98.91 |
| ATOM | 1.00 | 99.59 | 0.61 | 99.53 | 2.15 | **99.40** | 2.52 | 99.10 | 7.93 | 97.27 | 2.84 | 98.97 |
| DOE | 1.80 | 99.37 | **0.25** | 99.65 | 2.00 | 99.36 | 5.65 | 98.75 | 10.15 | 97.28 | 3.97 | 98.88 |
| POEM | 1.20 | 99.53 | 0.80 | 99.10 | **1.47** | 99.26 | 2.93 | 99.13 | 7.65 | 97.35 | 2.81 | 98.87 |
| DAL | **0.80** | **99.65** | 0.90 | 99.46 | 1.70 | 99.34 | **2.30** | 99.14 | **7.65** | **97.45** | **2.68** | **99.01** |
| CIFAR-100 | | | | | | | | | | | | |
| Using ID data only | | | | | | | | | | | | |
| MSP | 84.39 | 71.18 | 60.36 | 85.59 | 82.63 | 75.69 | 83.32 | 73.59 | 82.37 | 73.69 | 78.61 | 75.95 |
| Free Energy | 85.24 | 73.71 | 23.05 | 95.89 | 81.11 | 79.02 | 79.63 | 76.35 | 80.18 | 75.65 | 69.84 | 80.12 |
| ASH | 70.09 | 83.56 | 13.20 | 97.71 | 69.87 | 82.56 | 63.69 | 83.59 | 79.70 | 74.87 | 59.31 | 84.46 |
| Mahalanobis | 51.00 | 88.70 | 91.60 | 69.69 | 38.48 | 91.86 | 47.07 | 89.09 | 82.70 | 74.18 | 72.37 | 82.70 |
| KNN | 52.10 | 88.83 | 68.82 | 79.00 | 42.17 | 90.59 | 42.79 | 89.07 | 92.21 | 61.08 | 59.62 | 81.71 |
| KNN+ | 32.50 | 93.86 | 47.41 | 84.93 | 39.82 | 91.12 | 43.05 | 88.55 | 63.26 | 79.28 | 45.20 | 87.55 |
| CSI | 64.50 | 84.62 | 25.88 | 95.93 | 70.62 | 80.83 | 61.50 | 86.74 | 83.08 | 77.11 | 61.12 | 95.05 |
| VOS | 78.06 | 92.59 | 40.40 | 92.90 | 85.77 | 70.20 | 82.46 | 77.22 | 82.31 | 75.47 | 73.80 | 91.67 |
| Using ID data and auxiliary OOD data | | | | | | | | | | | | |
| OE | 46.73 | 90.54 | 16.30 | 96.98 | 47.97 | 88.43 | 50.39 | 88.27 | 54.30 | 87.11 | 43.14 | 90.27 |
| Energy-OE | 35.34 | 94.74 | 16.27 | 97.25 | 33.21 | 93.25 | 46.13 | 90.62 | 50.45 | 90.04 | 36.28 | 93.18 |
| ATOM | 24.80 | 95.15 | 17.83 | 96.76 | 47.83 | 91.06 | 44.86 | **91.80** | 53.92 | 88.88 | 37.84 | 92.73 |
| DOE | 43.10 | 91.83 | 13.95 | **97.56** | 47.25 | 87.88 | 49.40 | 88.62 | 51.05 | 88.08 | 40.95 | 90.79 |
| POEM | 22.27 | **96.28** | **13.66** | 97.52 | 42.46 | 91.97 | 45.94 | 90.42 | 49.50 | 90.21 | 34.77 | 93.28 |
| DAL | **19.35** | 96.21 | 16.05 | 96.78 | **26.05** | **94.23** | **37.60** | 91.57 | **49.35** | **90.81** | **29.68** | **93.92** |

where the model capacity is not enough to learn from all the auxiliary OOD data, deviating from our considered issue in OOD distribution discrepancy. Moreover, for the previous works that adopt similar concepts in the worst-case OOD learning, e.g., VOS and DOE, DAL also reveals better results, with 1.29 and 30.89 improvements on the CIFAR-10 dataset and 11.27 and 44.12 improvements on the CIFAR-100 dataset w.r.t. FPR95. It indicates that our theoretical-driven scheme can also guide the algorithm designs with practical effectiveness. Note that many previous works use advanced scoring strategies other than MSP, and thus our experiment above is not completely fair to us. Therefore, in Appendix F.3, we also combine DAL with many advanced scoring strategies other than MSP, which can further improve our performance.

## 5.2 Hard OOD Detection

We further consider hard OOD scenarios (Sun et al., 2022), of which the test OOD data are very similar to that of the ID cases. Following the common setup (Sun et al., 2022) with the CIFAR-10 dataset being the ID case, we evaluate our DAL on three hard OOD datasets, namely, LSUN-Fix (Yu et al., 2015), ImageNet-Resize (Deng et al., 2009), and CIFAR-100. Note that data in ImageNet-Resize (1000 classes) with the same semantic space as Tiny-ImageNet (200 classes) are removed. We select a set of strong baselines that

Table 2: Comparison between our method and advanced methods on hard OOD detection. ↓ (or ↑) indicates smaller (or larger) values are preferred, and a bold font indicates the best result in a column.

| Methods | LSUN-Fix | | ImageNet-Resize | | CIFAR-100 | |
|---|---|---|---|---|---|---|
| | FPR95↓ | AUROC↑ | FPR95↓ | AUROC↑ | FPR95↓ | AUROC↑ |
| Using ID data only | | | | | | |
| Free Energy | 6.42 | 98.85 | 46.46 | 89.02 | 50.47 | 87.08 |
| ASH | 4.00 | 98.20 | 46.18 | 88.85 | 54.31 | 83.71 |
| KNN+ | 24.88 | 95.75 | 30.52 | 94.85 | 40.00 | 89.11 |
| CSI | 39.79 | 93.63 | 37.47 | 93.93 | 45.64 | 87.64 |
| Using ID data and auxiliary OOD data | | | | | | |
| OE | 1.75 | 99.47 | 6.76 | 98.58 | 29.40 | 94.20 |
| DOE | 1.97 | 98.71 | 5.98 | 98.75 | 29.75 | 94.24 |
| POEM | **1.24** | 98.93 | 6.56 | 98.37 | 35.11 | 91.80 |
| DAL | 1.39 | **99.47** | **5.60** | **98.80** | **25.45** | **94.34** |

are competent in hard OOD detection, summarizing the experiments in Table 2. As we can see, our method can beat these advanced methods across the considered datasets, even for the challenging

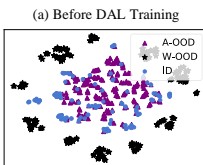
(a) Before DAL Training

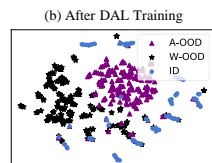
(b) After DAL Training

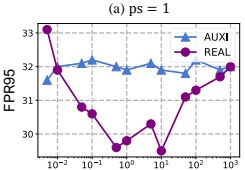
(a) ps = 1

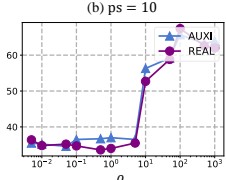
(b) ps = 10

Figure 2: Illustrations of embedding features for the ID, the auxiliary OOD (A-OOD), and the worst OOD (W-OOD) data. We adopt the t-SNE visualization on the CIFAR-10 dataset and illustrate the results before and after DAL training.

Figure 3: FPR95 curves across various $\rho$ on the CIFAR-100. We report both the results for the average real OOD data (REAL) and the auxiliary OOD data (AUXI), where we consider two hyperparameter setups, i.e., ps = 1 and ps = 10.

CIFAR-10 versus CIFAR-100 setting. The reason is that our distributional augmentation directly learns from OOD data close to ID pattern, which can cover hard OOD cases.

## 5.3 Ablation Study

We further conduct an ablation study to demonstrate two mechanisms that mainly contribute to our open-world effectiveness, namely, OOD data generation and Wasserstein ball constraint.

**OOD Data Generation.** DAL learns from the worst OOD data to mitigate the OOD distribution discrepancy. To understand such an OOD generation scheme, we employ the t-SNE visualization (Van der Maaten and Hinton, 2008) for the ID, the auxiliary OOD, and the worst OOD data. Figure 2 summarizes the results before and after DAL training. Before training, the ID and auxiliary OOD data overlap largely, indicating that the original model is not effective at distinguishing between them. Then, DAL does not directly train the model on auxiliary OOD data but instead perturbs it to further confuse the model beyond the overlap region. After DAL training, the overlap region between ID and auxiliary OOD data shrinks. Additionally, perturbing the original OOD data becomes more difficult, indicating that the model has learned to handle various worst-case OOD scenarios.

**Wasserstein Ball Constraint.** The choice of $\rho$ determines the radius of the Wasserstein ball. Larger values of $\rho$ reduce estimation error and improve model generalization, as stated in Theorem 3. However, larger values of $\rho$ also increase the approximate risk $\min_{\mathbf{w} \in \mathcal{W}} R_D(\mathbf{w}; \rho)$ as it becomes more challenging to ensure uniform model performance with increased distributional perturbation. Figure 3 shows the FPR95 curves on the CIFAR-100 dataset for both the real and the surrogate OOD data, revealing the trade-off in selecting $\rho$. Here, we consider two setups of $\rho$, i.e., ps = 1 (default) and ps = 10 (large perturbation strength). First, when the perturbation strength is very large (i.e., ps = 10), the model can easily fail for training if the value of $\rho$ is also large (e.g., $\rho = 100$), indicating that large value of $\rho$ can lead to a large approximation error. However, such an issue can be overcome by selecting a relatively small value of $\rho$ (e.g., $\rho = 0.5$).

## 6 Conclusion

Outlier exposure is one of the most powerful methods in OOD detection, but the discrepancy between the auxiliary and (unseen) real OOD data can hinder its practical effectiveness. To address such an issue, we have formalized it as the OOD distribution discrepancy and developed an effective learning framework to mitigate its negative impacts. Specifically, we consider a specific distribution set that contains all distributions in a Wasserstein ball centered on the auxiliary OOD distribution. Then, models trained over worst-case OOD data in the ball can ensure improved performance toward open-world OOD detection. Overall, as pioneers in critically analyzing the open-world setting with theoretical analysis, we are committed to raising attention to the OOD distribution discrepancy issue and encouraging further research in this direction.

## Acknowledgments and Disclosure of Funding

QZW, YGZ, and BH were supported by the NSFC Young Scientists Fund No. 62006202, NSFC General Program No. 62376235, Guangdong Basic and Applied Basic Research Foundation No. 2022A1515011652, HKBU Faculty Niche Research Areas No. RC-FNRA-IG/22-23/SCI/04, and HKBU CSD Departmental Incentive Scheme. FL was supported by Australian Research Council (ARC) under Award No. DP230101540, and by NSF and CSIRO Responsible AI Program under Award No. 2303037. YXL was supported by the AFOSR Young Investigator Program under award number FA9550-23-1-0184, National Science Foundation (NSF) Award No. IIS-2237037 & IIS-2331669, and Office of Naval Research under grant number N00014-23-1-2643.

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
