# A  Related Works

In this section, we discuss the related studies in OOD detection.

**OOD Detection Methods.** Existing works in OOD detection can be mainly categorized into three categories, namely, the post-hoc methods, the representation-based methods, and the outlier exposure. For the post-hoc methods, they believe a well-trained ID classifier can already lead to effective OOD detection (Hendrycks and Gimpel, 2017), given proper scoring functions to indicate ID and OOD cases. Existing scoring functions are built upon the classifiers, taking logit outputs (Hendrycks and Gimpel, 2017; Liang et al., 2018; Liu et al., 2020; Sun et al., 2021; Wang et al., 2021a; Lakshminarayanan et al., 2017; Wang et al., 2021a; Huang and Li, 2021), embedding features (Lee et al., 2018a; Sastry and Oore, 2020; Wang et al., 2022a; Lin et al., 2021; Sun et al., 2022; Morteza and Li, 2022; Luo et al., 2023), or gradient information (Huang et al., 2021; Liang et al., 2018; Igoe et al., 2022) as its inputs and returning a score value to indicate the confidence for an ID case. Recent works focus on adaptation strategies for specific tasks (Huang et al., 2021; Zhu et al., 2023a) and non-parametric approaches (Sun et al., 2022), which may motivate future works.

Other works believe that training procedures are indispensable in OOD detection. For representation-based methods, researchers assume that a good ID representation is all we need for effective OOD detection. Therein, researchers study contrastive learning methods (Sehwag et al., 2021; Wang et al., 2022b), data augmentation (Tack et al., 2020; Zheng et al., 2023), constraints on embedding features (Du et al., 2022b; Ming et al., 2023; Zaeemzadeh et al., 2021) or model output (Wei et al., 2022). However, some of the adopted scoring functions in representation-based methods are complex. It can make us overestimate the true effects of representation learning, which may require further studies. For outlier exposure, related methods directly make the model learn from OOD data with low OOD score predictions (Hendrycks et al., 2019; Liu et al., 2020; Huang et al., 2023a). Related works studies sampling strategies (Zhu et al., 2023b; Ming et al., 2022; Chen et al., 2021), adversarial robust learning (Li and Vasconcelos, 2020; Lee et al., 2018a; Hein et al., 2019), meta learning (Jeong and Kim, 2020), and regularization strategies (Van Amersfoort et al., 2020). Other works consider the situations where OOD data are inaccessible, studying various outlier synthesis strategies (Lee et al., 2018b; Vernekar et al., 2019; Du et al., 2022a; Tao et al., 2023). Although outlier exposure typically reveals promising results, the difference between auxiliary and real OOD data largely hinders its real-world detection power, similar to conclusions in domain adaptation (Luo et al., 2023, 2020).

**OOD Detection Theory.** Zhang et al. (2021) gives an explanation of why there exist OOD data that have higher probabilities or densities than the data from the ID distribution in the deep generative models. Zhang et al. (2021) understands OOD detection via goodness-of-fit tests and points out that OOD detection should be defined based on the data distribution's typical set if we hope OOD detection can be successful. Morteza and Li (2022) develops a novel unified framework that helps researchers to understand the theoretical connections among some representative OOD detection methods. Fang et al. (2021, 2022) develop the probably approximately correct (PAC) learning theory for OOD detection and gives a series of sufficient and necessary conditions for the PAC learnability of OOD detection. Fang et al. (2022) has proven that although OOD detection cannot be PAC learnable in the distribution-free case, OOD detection can be successful in many practical scenarios. Note that Zhang et al. (2021), Morteza and Li (2022), and Fang et al. (2022) all consider the case that the auxiliary OOD data are unavailable. Therefore, to ensure the leanability of OOD detection, some strong conditions are necessary (Zhang et al., 2021; Fang et al., 2022). To explore the outlier exposure case in OOD detection, Bitterwolf et al. (2022) shows that several representative OOD detection methods that optimize an objective that includes predictions on auxiliary OOD data are equivalent to the binary discriminator. Compared to Zhang et al. (2021); Morteza and Li (2022); Fang et al. (2022), our paper mainly focuses on the case that the auxiliary OOD data are available. Using the auxiliary OOD data, we can weaken the strong conditions proposed by Fang et al. (2022) and provide more reasonable and practical learning bounds for OOD detection. Compared to Bitterwolf et al. (2022), our theory mainly focuses on the learnability of OOD detection in the outlier exposure case and provides theoretical support to our practical method.

# B Notations

In this section, we summarize the important notations in Table 3.

Table 3: Main notations and their descriptions.

| Notation | Description |
|---|---|
| **Spaces** | |
| $\mathcal{X}$ and $\mathcal{Y}$ | the feature space and the ID label space $\{1, \ldots, C\}$ |
| $\mathcal{W}$ | the parameter space |
| $\mathfrak{D}$ | the distribution space |
| $\mathcal{E}$ | the embedding space |
| **Distributions** | |
| $X_\mathrm{I}, X_\mathrm{A}, X_\mathrm{O}$ | ID feature, auxiliary OOD feature, and real OOD feature |
| $Y_\mathrm{I}$ and $Y_\mathrm{O}$ | ID label and OOD label random variable |
| $D_{X_\mathrm{I}Y_\mathrm{I}}$ and $D_{X_\mathrm{O}Y_\mathrm{O}}$ | ID joint distribution and OOD joint distribution |
| $D_{X_\mathrm{A}}$ | the auxiliary OOD distribution |
| $\delta$ | the dirac measure |
| **Data and Models** | |
| $S$ and $T$ | ID training data and auxiliary OOD training data |
| $n$ and $m$ | the number of ID data and the number of auxiliary OOD data |
| $\mathbf{x}_\mathrm{I}$ and $\mathbf{x}_\mathrm{A}$ | ID data and auxiliary OOD data |
| $y_\mathrm{I}$ | ID label |
| $\mathbf{f_w}$ | the model: $\mathcal{X} \to \mathbb{R}^C$, parameterized by $\mathbf{w} \in \mathcal{W}$ |
| $\mathbf{e}$ and $\mathbf{h}$ | the feature extractor and the classifier |
| $s(\cdot; \mathbf{f})$ | the scoring function: $\mathcal{X} \to \mathbb{R}$ |
| $g_\lambda(\cdot)$ | the OOD detector: $\mathcal{X} \to \{\mathrm{ID}, \mathrm{OOD}\}$, with threshold $\lambda$ |
| **Distances** | |
| $c(\cdot, \cdot)$ | the cost function: $\mathcal{X} \times \mathcal{X} \to \mathbb{R}_+$ |
| $d(\cdot, \cdot)$ | the distance between two distributions: $\mathfrak{D} \times \mathfrak{D} \to \mathbb{R}_+$ |
| $\mathrm{W}_c$ | the Wasserstein-1 distance: $\mathfrak{D} \times \mathfrak{D} \to \mathbb{R}_+$ |
| $\rho$ | the radius of the Wasserstein ball |
| $\|\cdot\|_p$ | $l_p$ norm |
| **Loss and Risk** | |
| $\ell$ and $\ell_\mathrm{OE}$ | ID loss function and OOD loss function |
| $R_\mathrm{I}(\mathbf{w})$ and $\widehat{R}_\mathrm{I}(\mathbf{w})$ | the expected risk and the empirical risk corresponding to $D_{X_\mathrm{I}Y_\mathrm{I}}$ |
| $R_\mathrm{A}(\mathbf{w})$ and $\widehat{R}_\mathrm{A}(\mathbf{w})$ | the expected risk and the empirical risk corresponding to $D_{X_\mathrm{A}}$ |
| $R_\mathrm{O}(\mathbf{w})$ | the expected risk corresponding to $D_{X_\mathrm{O}}$ |
| $R_D(\mathbf{w})$ | the real detection risk corresponding to $D$ |
| $\phi_\gamma(\mathbf{w}; \mathbf{x})$ | the surrogate function |
| $R_\mathrm{O}(\mathbf{w}; \rho)$ and $\widehat{R}_\mathrm{O}(\mathbf{w}; \rho)$ | the expected DAL risk and the empirical one corresponding to $D_{X_\mathrm{A}}$ |
| $R_D(\mathbf{w}; \rho)$ and $\widehat{R}_D(\mathbf{w}; \rho)$ | the expected DAL risk and the empirical one corresponding to $D$ |
| **Hypothesis Space** | |
| $\mathcal{F}$ and $\mathcal{F}_\mathrm{OE}$ | the model classes with respect to $\ell$ and $\ell_\mathrm{OE}$ |
| $\mathcal{N}(\cdot, \epsilon, L^\infty)$ | the covering number |

# C Proofs of Theorems

We provide the detailed proofs for our theoretical results in Sections 4.1.

## C.1 Covering Number

We use the covering number for the model classes in our derivation. Here, we give the formal definition.

**Definition 2** ($\epsilon$-covering (Vershynin, 2018)). Let $(V, \|\cdot\|)$ be a normed space, $\Theta \subset V$, and $B(\cdot, \epsilon)$ the ball of radius $\epsilon$. Then $\{V_1, \ldots, V_N\}$ is an $\epsilon$-covering of $\Theta$ if $\Theta \subset \bigcup_{i=1}^{N} B(V_i, \epsilon)$, or equivalently, $\forall \theta \in \Theta$, $\exists i$ such that $\|\theta - V_i\| \leq \epsilon$.

Upon our definition of $\epsilon$-covering, the covering number is the minimal number of $\epsilon$-balls one needs to cover $\Theta$.

**Definition 3** (Covering Number (Vershynin, 2018)).

$$\mathcal{N}(\Theta, \|\cdot\|, \epsilon) = \min\{n : \exists \ \epsilon\text{-covering over } \Theta \text{ of size } n\}.$$

## C.2 Proof of Lemma 1

*Proof of Lemma 1.* Because of $D_{X_A} \in \mathfrak{D}$, according to the definite of infimum, it is clear that

$$\inf_{D_{X'} \in \mathfrak{D}} d(D_{X'}, D_{X_O}) \leq d(D_{X_A}, D_{X_O}).$$

To prove the second result, we consider a special distribution $D'$, which is defined as follows: for some $u \in [0, 1]$,

$$D' = (1 - u)D_{X_O} + uD_{X_A}.$$

Because $c(\cdot, \cdot)$ is a continuous metric, Kantorovich–Rubinstein duality (Villani, 2021) implies that

$$\begin{aligned}
\mathrm{W}_c(D', D_{X_O}) &= \sup_{\|f\|_{\mathrm{Lip}} \leq 1} \int_{\mathcal{X}} f(\mathbf{x}) \mathrm{d}D'(\mathbf{x}) - \int_{\mathcal{X}} f(\mathbf{x}) \mathrm{d}D_{X_O}(\mathbf{x}) \\
&= u \sup_{\|f\|_{\mathrm{Lip}} \leq 1} \int_{\mathcal{X}} f(\mathbf{x}) \mathrm{d}D_{X_A}(\mathbf{x}) - \int_{\mathcal{X}} f(\mathbf{x}) \mathrm{d}D_{X_O}(\mathbf{x}) \\
&= u\mathrm{W}_c(D_{X_A}, D_{X_O})
\end{aligned}$$

Similarly, we can obtain that

$$\mathrm{W}_c(D', D_{X_A}) = (1 - u)\mathrm{W}_c(D_{X_A}, D_{X_O})$$

**Case 1.** If $\mathrm{W}_c(D_{X_A}, D_{X_O}) \leq \rho$, then it is clear that

$$\inf_{D_{X'} \in \mathfrak{D}} \mathrm{W}_c(D_{X'}, D_{X_O}) \leq \max\{\mathrm{W}_c(D_{X_A}, D_{X_O}) - \rho, 0\}.$$

**Case 2.** If $\mathrm{W}_c(D_{X_A}, D_{X_O}) > \rho$, then we set $u = 1 - \rho/\mathrm{W}_c(D_{X_A}, D_{X_O})$. Therefore,

$$\mathrm{W}_c(D', D_{X_A}) = \rho, \qquad \mathrm{W}_c(D', D_{X_O}) = \mathrm{W}_c(D_{X_A}, D_{X_O}) - \rho,$$

which implies that

$$\inf_{D_{X'} \in \mathfrak{D}} \mathrm{W}_c(D_{X'}, D_{X_O}) \leq \mathrm{W}_c(D', D_{X_O}) = \mathrm{W}_c(D_{X_A}, D_{X_O}) - \rho \leq \max\{\mathrm{W}_c(D_{X_A}, D_{X_O}) - \rho, 0\}.$$

We have completed this proof by combining Cases 1 and 2. $\square$

## C.3 Proof of Theorem 1

*Proof of Theorem 1.* One can find a similar proof from Blanchet et al. (2019); Blanchet and KarthyekRajhaaA. (2016); Sinha et al. (2018). We omit it here. $\square$

## C.4 Proof of Theorem 2

*Proof of Theorem 2.* We first recall the notations as follows:

$$R_{\mathrm{I}}(\mathbf{w}) = \mathbb{E}_{(\mathbf{x},y) \sim D_{X_{\mathrm{I}} Y_{\mathrm{I}}}} \ell(\mathbf{f}_{\mathbf{w}}; \mathbf{x}, y),$$

$$\widehat{R}_{\mathrm{I}}(\mathbf{w}) = \frac{1}{n} \sum_{i=1}^{n} \ell(\mathbf{f}_{\mathbf{w}}; \mathbf{x}_{\mathrm{I}}^i, y_{\mathrm{I}}^i),$$

$$R_{\mathrm{O}}(\mathbf{w}; \rho) = \sup_{\mathrm{W}_c(D_{X'}, D_{X_{\mathrm{A}}}) \leq \rho} \mathbb{E}_{\mathbf{x} \sim D_{X'}} \ell_{\mathrm{OE}}(\mathbf{f}_{\mathbf{w}}; \mathbf{x}),$$

$$\widehat{R}_{\mathrm{O}}(\mathbf{w}; \rho) = \sup_{\mathrm{W}_c(D_{X'}, \widehat{D}_{X_{\mathrm{A}}}) \leq \rho} \mathbb{E}_{\mathbf{x} \sim D_{X'}} \ell_{\mathrm{OE}}(\mathbf{f}_{\mathbf{w}}; \mathbf{x}).$$

Let $\mathbf{w}^*$ be the solution of $\min_{\mathbf{w} \in \mathcal{W}} R_D(\mathbf{w}; \rho)$. Then

$$R_D(\widehat{\mathbf{w}}; \rho) - R_D(\mathbf{w}^*; \rho)$$
$$\leq R_D(\widehat{\mathbf{w}}; \rho) - \widehat{R}_D(\widehat{\mathbf{w}}; \rho) + \widehat{R}_D(\widehat{\mathbf{w}}; \rho) - R_D(\mathbf{w}^*; \rho) + \widehat{R}_D(\mathbf{w}^*; \rho) - \widehat{R}_D(\mathbf{w}^*; \rho)$$
$$\leq [R_{\mathrm{I}}(\widehat{\mathbf{w}}) - R_{\mathrm{I}}(\mathbf{w}^*)] + \alpha[R_{\mathrm{O}}(\widehat{\mathbf{w}}; \rho) - R_{\mathrm{O}}(\mathbf{w}^*; \rho)] - [\widehat{R}_{\mathrm{I}}(\widehat{\mathbf{w}}) - \widehat{R}_{\mathrm{I}}(\mathbf{w}^*)] - \alpha[\widehat{R}_{\mathrm{O}}(\widehat{\mathbf{w}}; \rho) - \widehat{R}_{\mathrm{O}}(\mathbf{w}^*; \rho)]$$
$$= [R_{\mathrm{I}}(\widehat{\mathbf{w}}) - \widehat{R}_{\mathrm{I}}(\widehat{\mathbf{w}})] + \alpha[R_{\mathrm{O}}(\widehat{\mathbf{w}}; \rho) - \widehat{R}_{\mathrm{O}}(\widehat{\mathbf{w}}; \rho)] - [R_{\mathrm{I}}(\mathbf{w}^*) - \widehat{R}_{\mathrm{I}}(\mathbf{w}^*)] - \alpha[R_{\mathrm{O}}(\mathbf{w}^*; \rho) - \widehat{R}_{\mathrm{O}}(\mathbf{w}^*; \rho)]. \tag{12}$$

By Lemmas 4 and 9, we have that with the probability at least $1 - 2e^{-t} > 0$, for any $\rho > 0$,

$$[R_{\mathrm{I}}(\widehat{\mathbf{w}}) - \widehat{R}_{\mathrm{I}}(\widehat{\mathbf{w}})] + \alpha[R_{\mathrm{O}}(\widehat{\mathbf{w}}; \rho) - \widehat{R}_{\mathrm{O}}(\widehat{\mathbf{w}}; \rho)]$$
$$\leq \frac{b_0 M_\ell}{\sqrt{n}} \int_0^1 \sqrt{\log \mathcal{N}(\mathcal{F}, M_\ell \epsilon, L^\infty)} d\epsilon \tag{13}$$
$$+ \alpha b_1 \sqrt{\frac{M_{\ell_{\mathrm{OE}}}^3}{\rho^2 m}} \int_0^1 \sqrt{\log \mathcal{N}(\mathcal{F}_{\mathrm{OE}}, M_{\ell_{\mathrm{OE}}} \epsilon, L^\infty)} d\epsilon + \alpha b_2 M_{\ell_{\mathrm{OE}}} \sqrt{\frac{2t}{m}} + M_\ell \sqrt{\frac{2t}{n}}.$$

By Lemmas 8 and 10, we have that with the probability at least $1 - 2e^{-t} > 0$, for any $\rho > 0$,

$$[R_{\mathrm{I}}(\mathbf{w}^*) - \widehat{R}_{\mathrm{I}}(\mathbf{w}^*)] + \alpha[R_{\mathrm{O}}(\mathbf{w}^*; \rho) - \widehat{R}_{\mathrm{O}}(\mathbf{w}^*; \rho)] \leq M_\ell \sqrt{\frac{2t}{n}} + 2\alpha M_{\ell_{\mathrm{OE}}} \sqrt{\frac{2t}{m}}. \tag{14}$$

Combining Eqs. (12), (13) and (14), we have that with the probability at least $1 - 4e^{-t} > 0$, for any $\rho > 0$,

$$R_D(\widehat{\mathbf{w}}; \rho) - R_D(\mathbf{w}^*; \rho)$$
$$\leq \frac{b_0 M_\ell}{\sqrt{n}} \int_0^1 \sqrt{\log \mathcal{N}(\mathcal{F}, M_\ell \epsilon, L^\infty)} d\epsilon$$
$$+ \alpha b_1 \sqrt{\frac{M_{\ell_{\mathrm{OE}}}^3}{\rho^2 m}} \int_0^1 \sqrt{\log \mathcal{N}(\mathcal{F}_{\mathrm{OE}}, M_{\ell_{\mathrm{OE}}} \epsilon, L^\infty)} d\epsilon + 2 M_\ell \sqrt{\frac{2t}{n}} + b_2 \alpha M_{\ell_{\mathrm{OE}}} \sqrt{\frac{2t}{m}},$$

where $b_0$, $b_1$ and $b_2$ are uniform constants. $\qquad \square$

*Remark* 4. One can prove that the cross-entropy and the exponential losses are bounded and lipschitz w.r.t. $\mathbf{w}$ for deep models with softmax outputs (Golowich et al., 2018), if

- activation functions are 1-Lipschitz;
- inputs are from bounded feature space $\mathcal{X}$;
- the parameter space $\mathcal{W}$ is bounded (e.g., with regularization).

More specifically, when the F-norm bounds parameters, the softmax output is continuous and never attains infinity. If we further assume that inputs are from a bounded feature space, then the model is a continuous function over the bounded space, implying that model outputs have upper and lower bounds. Thus, the cross-entropy and the exponential loss can be bounded in practice, and our assumptions are practical.

### C.5 Corollary 1

**Corollary 1.** *Given the same conditions in Theorem 2, if*

- $\ell_{\mathrm{OE}}(\mathbf{f}_{\mathbf{w}};\mathbf{x})$ *is $L_{\mathrm{OE}}$-Lipschitz w.r.t. norm $\|\cdot\|$, i.e.,*

$$|\ell_{\mathrm{OE}}(\mathbf{f}_{\mathbf{w}};\mathbf{x}) - \ell_{\mathrm{OE}}(\mathbf{f}_{\mathbf{w}'};\mathbf{x})| \leq L_{\mathrm{OE}}\|\mathbf{w} - \mathbf{w}'\|,$$

- $\ell_{\mathrm{OE}}(\mathbf{f}_{\mathbf{w}};\mathbf{x})$ *is $L_c$-Lipschitz w.r.t. $c(\cdot,\cdot)$, i.e.,*

$$|\ell_{\mathrm{OE}}(\mathbf{f}_{\mathbf{w}};\mathbf{x}) - \ell_{\mathrm{OE}}(\mathbf{f}_{\mathbf{w}};\mathbf{x}')| \leq L_c c(\mathbf{x},\mathbf{x}'),$$

- $\ell(\mathbf{f}_{\mathbf{w}};\mathbf{x},y)$ *is $L$-Lipschitz w.r.t. norm $\|\cdot\|$, i.e.,*

$$|\ell(\mathbf{f}_{\mathbf{w}};\mathbf{x},y) - \ell(\mathbf{f}_{\mathbf{w}'};\mathbf{x},y)| \leq L\|\mathbf{w} - \mathbf{w}'\|,$$

- *the parameter space $\mathcal{W} \subset \mathbb{R}^{d'}$ satisfies that*

$$\mathrm{diam}(\mathcal{W}) = \sup_{\mathbf{w},\mathbf{w}'\in\mathcal{W}} \|\mathbf{w} - \mathbf{w}'\| < +\infty.$$

*Let $\widehat{\mathbf{w}}$ be the optimal solution of Eq. (8), i.e.,*

$$\widehat{\mathbf{w}} \in \arg\min_{\mathbf{w}\in\mathcal{W}} \widehat{R}_D(\mathbf{w};\rho).$$

*With the probability at least $1 - 4e^{-t} > 0$, for any $\rho > 0$,*

$$R_D(\widehat{\mathbf{w}};\rho) - \min_{\mathbf{w}\in\mathcal{W}} R_D(\mathbf{w};\rho) \leq \tilde{\epsilon}(n,m;t),$$

*where*

$$
\begin{aligned}
\tilde{\epsilon}(n,m;t) =& b_0\sqrt{\frac{M_\ell \mathrm{diam}(\mathcal{W})Ld'}{n}} \\
&+ \alpha b_1 \min\{L_c, \frac{M_{\ell_{\mathrm{OE}}}}{\rho}\}\sqrt{\frac{\mathrm{diam}(\mathcal{W})L_{\mathrm{OE}}d'}{m}} \\
&+ 2M_\ell\sqrt{\frac{2t}{n}} + \alpha b_2 M_{\ell_{\mathrm{OE}}}\sqrt{\frac{2t}{m}},
\end{aligned}
$$

*here $b_0$, $b_1$ and $b_2$ are uniform constants.*

*Proof of Corollary 1.* By Lemmas 7 and 11, we have that with the probability at least $1 - 2e^{-t} > 0$, for any $\rho > 0$,

$$
\begin{aligned}
&[R_{\mathrm{I}}(\widehat{\mathbf{w}}) - \widehat{R}_{\mathrm{I}}(\widehat{\mathbf{w}})] + \alpha[R_{\mathrm{O}}(\widehat{\mathbf{w}};\rho) - \widehat{R}_{\mathrm{O}}(\widehat{\mathbf{w}};\rho)] \\
&\leq \left[b_0\sqrt{\frac{M_\ell \mathrm{diam}(\mathcal{W})Ld'}{n}} + M_\ell\sqrt{\frac{2t}{n}}\right] + \alpha b_1 \min\{L_c, \frac{M_{\ell_{\mathrm{OE}}}}{\rho}\}\sqrt{\frac{\mathrm{diam}(\mathcal{W})L_{\mathrm{OE}}d'}{m}} + \alpha b_2 M_{\ell_{\mathrm{OE}}}\sqrt{\frac{t}{m}}
\end{aligned}
$$
(15)

By Lemmas 8 and 10, we have that with the probability at least $1 - 2e^{-t} > 0$, for any $\rho > 0$,

$$[R_{\mathrm{I}}(\mathbf{w}^*) - \widehat{R}_{\mathrm{I}}(\mathbf{w}^*)] + \alpha[R_{\mathrm{O}}(\mathbf{w}^*;\rho) - \widehat{R}_{\mathrm{O}}(\mathbf{w}^*;\rho)] \leq M_\ell\sqrt{\frac{2t}{n}} + 2\alpha M_{\ell_{\mathrm{OE}}}\sqrt{\frac{2t}{m}}. \tag{16}$$

Using Eqs. (15), (16) and Eq. (12), we know that with the probability at least $1 - 4e^{-t} > 0$, for any $\rho > 0$,

$$
\begin{aligned}
&R_D(\widehat{\mathbf{w}};\rho) - \min_{\mathbf{w}\in\mathcal{W}} R_D(\mathbf{w};\rho) \\
&\leq b_0\sqrt{\frac{M_\ell \mathrm{diam}(\mathcal{W})Ld'}{n}} + \alpha b_1 \min\{L_c, \frac{M_{\ell_{\mathrm{OE}}}}{\rho}\}\sqrt{\frac{\mathrm{diam}(\mathcal{W})L_{\mathrm{OE}}d'}{m}} + 2M_\ell\sqrt{\frac{2t}{n}} + \alpha b_2 M_{\ell_{\mathrm{OE}}}\sqrt{\frac{2t}{m}},
\end{aligned}
$$

where $b_0$, $b_1$ and $b_2$ are uniform constants. $\qquad\square$

## C.6 Proof of Theorem 3

*Proof of Theorem 3.* Consider
$$R_D(\widehat{\mathbf{w}}) - R_D(\widehat{\mathbf{w}}; \rho).$$

It is clear that
$$R_D(\widehat{\mathbf{w}}) - R_D(\widehat{\mathbf{w}}; \rho) = \alpha[R_O(\widehat{\mathbf{w}}) - R_O(\widehat{\mathbf{w}}; \rho)].$$

Let $D' = (1 - u)D_{X_O} + uD_{X_A}$ and
$$\rho_O = W_c(D_{X_O}, D_{X_A}).$$

Because $c(\cdot, \cdot)$ is a continuous metric, Kantorovich–Rubinstein duality Villani (2021) implies that

$$\begin{aligned}
&W_c(D', D_{X_A}) \\
&= \sup_{\|f\|_{\mathrm{Lip}} \leq 1} \int_{\mathcal{X}} f(\mathbf{x})\mathrm{d}D'(\mathbf{x}) - \int_{\mathcal{X}} f(\mathbf{x})\mathrm{d}D_{X_A}(\mathbf{x}) \\
&= (1 - u) \sup_{\|f\|_{\mathrm{Lip}} \leq 1} \int_{\mathcal{X}} f(\mathbf{x})\mathrm{d}D_{X_O}(\mathbf{x}) - \int_{\mathcal{X}} f(\mathbf{x})\mathrm{d}D_{X_A}(\mathbf{x}) \\
&= (1 - u)W_c(D_{X_O}, D_{X_A}) \\
&= (1 - u)\rho_O
\end{aligned}$$

Let
$$u = 1 - \frac{\rho}{\rho_O}.$$

**Case 1.** If $\rho \geq \rho_O$, then
$$R_D(\widehat{\mathbf{w}}) \leq R_D(\widehat{\mathbf{w}}; \rho)$$

**Case 2.** If $\rho < \rho_O$, then by Lemma 6
$$R_O(\widehat{\mathbf{w}}) - R_O(\widehat{\mathbf{w}}; \rho) \leq R_O(\widehat{\mathbf{w}}) - \mathbb{E}_{\mathbf{x} \sim D'}\ell_{\mathrm{OE}}(\mathbf{f_w}; \mathbf{x}) \leq L_c(\rho_O - \rho),$$

By Cases 1 and 2, we have shown that
$$R_D(\widehat{\mathbf{w}}) - R_D(\widehat{\mathbf{w}}; \rho) \leq \alpha L_c \max\{W_c(D_{X_o}, D_{X_A}) - \rho, 0\}.$$

Then by Theorem 2, we complete this proof. $\quad\square$

# D   Necessary Lemmas

**Lemma 2.** *Assume that*

- $|\ell_{\mathrm{OE}}(\mathbf{f_w};\mathbf{x})| \leq M_{\ell_{\mathrm{OE}}}$,

- $c: \mathcal{X} \times \mathcal{X} \to \mathbb{R}_+$ *is a continuous metric,*

*then*

- $|\phi_\gamma(\mathbf{w};\mathbf{x})| \leq M_{\ell_{\mathrm{OE}}}$,

- *for some $\mathbf{w}_0 \in \mathcal{W}$ and $\epsilon > 0$, when $\gamma_\epsilon$ ($\gamma_\epsilon \geq 0$) satisfies the following condition:*

$$\gamma_\epsilon \rho + \mathbb{E}_{\mathbf{x}\sim D}\phi_{\gamma_\epsilon}(\mathbf{w}_0;\mathbf{x}) \leq \inf_{\gamma\geq 0}[\gamma\rho + \mathbb{E}_{\mathbf{x}\sim D}\phi_\gamma(\mathbf{w}_0;\mathbf{x})] + \epsilon,$$

*then*

$$\gamma_\epsilon \leq \frac{2M_{\ell_{\mathrm{OE}}} + \epsilon}{\rho}.$$

*Proof of Lemma 2.* **First**, we prove: $|\phi_\gamma(\mathbf{w};\mathbf{x})| \leq M_{\ell_{\mathrm{OE}}}$.

Because $\phi_\gamma(\mathbf{w};\mathbf{x}) = \sup_{\mathbf{x}'\in\mathcal{X}}\{\ell_{\mathrm{OE}}(\mathbf{f_w};\mathbf{x}') - \gamma c(\mathbf{x}',\mathbf{x})\}$ and $c(\mathbf{x},\mathbf{x}') \geq 0$, it is clear that

$$\phi_\gamma(\mathbf{w};\mathbf{x}) \leq \sup_{\mathbf{x}'\in\mathcal{X}} \ell_{\mathrm{OE}}(\mathbf{f_w};\mathbf{x}') \leq M_{\ell_{\mathrm{OE}}}.$$

In addition, because $c(\mathbf{x},\mathbf{x}) = 0$, then

$$\phi_\gamma(\mathbf{w};\mathbf{x}) \geq \ell_{\mathrm{OE}}(\mathbf{w};\mathbf{x}) \geq -M_{\ell_{\mathrm{OE}}}.$$

Above inequalities have indicated that

$$|\phi_\gamma(\mathbf{w};\mathbf{x})| \leq M_{\ell_{\mathrm{OE}}}.$$

**Second**, we prove that

$$\gamma_\epsilon \leq \frac{2M_{\ell_{\mathrm{OE}}} + \epsilon}{\rho}.$$

By the dual theorem in Blanchet et al. (2019); Blanchet and KarthyekRajhaaA. (2016); Sinha et al. (2018), we can obtain that

$$\inf_{\gamma\geq 0}[\gamma\rho + \mathbb{E}_{\mathbf{x}\sim D}\phi_\gamma(\mathbf{w}_0;\mathbf{x})] = \sup_{\mathrm{W}_c(D',D)\leq\rho} \mathbb{E}_{\mathbf{x}\sim D'}\ell_{\mathrm{OE}}(\mathbf{f}_{\mathbf{w}_0};\mathbf{x}) \leq M_{\ell_{\mathrm{OE}}},$$

which implies that

$$\gamma_\epsilon \rho \leq M_{\ell_{\mathrm{OE}}} + \epsilon - \mathbb{E}_{\mathbf{x}\sim D}\phi_\gamma(\mathbf{w}_0;\mathbf{x}) \leq 2M_{\ell_{\mathrm{OE}}} + \epsilon.$$

Therefore,

$$\gamma_\epsilon \leq \frac{2M_{\ell_{\mathrm{OE}}} + \epsilon}{\rho}.$$

$\square$

**Lemma 3** (Theorem 3 in Sinha et al. (2018)). *Given the same assumptions in Theorem 2, then with the probability at least $1 - e^{-t} > 0$, for any $\gamma \geq 0$, $\rho \geq 0$ and $\mathbf{w} \in \mathcal{W}$,*

$$\sup_{\mathrm{W}_c(D_{X'},D_{X_A})\leq\rho} \mathbb{E}_{\mathbf{x}\sim D_{X'}}\ell_{\mathrm{OE}}(\mathbf{f_w};\mathbf{x})$$

$$\leq \gamma\rho + \frac{1}{m}\sum_{i=1}^m \phi_\gamma(\mathbf{w};\mathbf{x}_A^i) + b_2 M_{\ell_{\mathrm{OE}}}\sqrt{\frac{t}{m}} + \gamma b_1\sqrt{\frac{M_{\ell_{\mathrm{OE}}}}{m}}\int_0^1 \sqrt{\log\mathcal{N}(\mathcal{F}_{\mathrm{OE}}, M_{\ell_{\mathrm{OE}}}\epsilon, L^\infty)}\mathrm{d}\epsilon,$$

*where $b_1$ and $b_2$ are uniform constants.*

*Proof of Lemma 3.* This lemma is following Theorem 3 in Sinha et al. (2018). $\square$

**Lemma 4.** *Given the same assumptions in Theorem 2, then with the probability at least $1 - e^{-t} > 0$, for any $\rho > 0$ and $\mathbf{w} \in \mathcal{W}$, we have*

$$\sup_{\mathrm{W}_c(D_{X'}, D_{X_\mathrm{A}}) \le \rho} \mathbb{E}_{\mathbf{x} \sim D_\mathrm{O}} \ell_\mathrm{OE}(\mathbf{f_w}; \mathbf{x})$$

$$\le \sup_{\mathrm{W}_c(D_{X'}, \widehat{D}_{X_\mathrm{A}}) \le \rho} \mathbb{E}_{\mathbf{x} \sim D_\mathrm{O}} \ell_\mathrm{OE}(\mathbf{f_w}; \mathbf{x}) + b_2 M_{\ell_\mathrm{OE}} \sqrt{\frac{t}{m}} + b_1 \sqrt{\frac{M_{\ell_\mathrm{OE}}^3}{\rho^2 m}} \int_0^1 \sqrt{\log \mathcal{N}(\mathcal{F}_\mathrm{OE}, M_{\ell_\mathrm{OE}} \epsilon, L^\infty)} \mathrm{d}\epsilon,$$

*where $b_1$ and $b_2$ are uniform constants.*

*Proof of Lemma 4.* By Lemma 3, we know that with the probability at least $1 - e^{-t} > 0$, for any $\frac{3M_{\ell_\mathrm{OE}}}{\rho} \ge \gamma \ge 0$, $\rho \ge 0$ and $\mathbf{w} \in \mathcal{W}$, we have

$$\sup_{\mathrm{W}_c(D_{X'}, D_{X_\mathrm{A}}) \le \rho} \mathbb{E}_{\mathbf{x} \sim D_{X'}} \ell_\mathrm{OE}(\mathbf{f_w}; \mathbf{x})$$

$$\le \gamma \rho + \frac{1}{m} \sum_{i=1}^m \phi_\gamma(\mathbf{w}; \mathbf{x}_\mathrm{A}^i) + b_2 M_{\ell_\mathrm{OE}} \sqrt{\frac{t}{m}} + b_1 \sqrt{\frac{M_{\ell_\mathrm{OE}}^3}{\rho^2 m}} \int_0^1 \sqrt{\log \mathcal{N}(\mathcal{F}_\mathrm{OE}, M_{\ell_\mathrm{OE}} \epsilon, L^\infty)} \mathrm{d}\epsilon,$$

where $b_1$ and $b_2$ are uniform constants.

The above bound and Lemma 2 imply that with the probability at least $1 - e^{-t} > 0$, for any $\rho \ge 0$ and $\mathbf{w} \in \mathcal{W}$, we have

$$\sup_{\mathrm{W}_c(D_{X'}, D_{X_\mathrm{A}}) \le \rho} \mathbb{E}_{\mathbf{x} \sim D_{X'}} \ell_\mathrm{OE}(\mathbf{f_w}; \mathbf{x})$$

$$\le \inf_{\gamma \ge 0} \left( \gamma \rho + \frac{1}{m} \sum_{i=1}^m \phi_\gamma(\mathbf{w}; \mathbf{x}_\mathrm{A}^i) \right) + b_2 M_{\ell_\mathrm{OE}} \sqrt{\frac{t}{m}} + b_1 \sqrt{\frac{M_{\ell_\mathrm{OE}}^3}{\rho^2 m}} \int_0^1 \sqrt{\log \mathcal{N}(\mathcal{F}_\mathrm{OE}, M_{\ell_\mathrm{OE}} \epsilon, L^\infty)} \mathrm{d}\epsilon.$$

Combining the above inequality with the following equation:

$$\inf_{\gamma \ge 0} \left( \gamma \rho + \frac{1}{m} \sum_{i=1}^m \phi_\gamma(\mathbf{w}; \mathbf{x}_\mathrm{A}^i) \right) = \sup_{\mathrm{W}_c(D_{X'}, \widehat{D}_{X_\mathrm{A}}) \le \rho} \mathbb{E}_{\mathbf{x} \sim D_{X'}} \ell_\mathrm{OE}(\mathbf{f_w}; \mathbf{x}),$$

we complete this proof. $\qquad \square$

**Lemma 5.** *Given the same assumptions in Lemma 4, if $\ell_\mathrm{OE}(\mathbf{f_w}; \mathbf{x})$ is a $L_\mathrm{OE}$-Lipschitz function w.r.t. norm $\| \cdot \|$ for all $\mathbf{x} \in \mathcal{X}$ and the parameter space $\mathcal{W} \subset \mathbb{R}^{d'}$ satisfies that $\mathrm{diam}(\mathcal{W}) = \sup_{\mathbf{w}, \mathbf{w}' \in \mathcal{W}} \| \mathbf{w} - \mathbf{w}' \| < +\infty$, then with the probability at least $1 - e^{-t} > 0$, for any $\rho > 0$ and $\mathbf{w} \in \mathcal{W}$,*

$$\sup_{\mathrm{W}_c(D_{X'}, D_{X_\mathrm{A}}) \le \rho} \mathbb{E}_{\mathbf{x} \sim D_\mathrm{O}} \ell_\mathrm{OE}(\mathbf{f_w}; \mathbf{x})$$

$$\le \sup_{\mathrm{W}_c(D_{X'}, \widehat{D}_{X_\mathrm{A}}) \le \rho} \mathbb{E}_{\mathbf{x} \sim D_\mathrm{O}} \ell_\mathrm{OE}(\mathbf{f_w}; \mathbf{x}) + b_2 M_{\ell_\mathrm{OE}} \sqrt{\frac{t}{m}} + b_1 M_{\ell_\mathrm{OE}} \sqrt{\frac{\mathrm{diam}(\mathcal{W}) L_\mathrm{OE} d'}{\rho^2 m}},$$

*where $b_1$ and $b_2$ are uniform constants.*

*Proof.* The proof is similar to Corollary 1 in Sinha et al. (2018). Note that

$$\mathcal{F}_\mathrm{OE} = \{ \ell_\mathrm{OE}(\mathbf{f_w}; \mathbf{x}) : \mathbf{w} \in \mathcal{W} \},$$

and $\ell_\mathrm{OE}(\mathbf{f_w}; \mathbf{x})$ is $L_\mathrm{OE}$-Lipschitz w.r.t. norm $\| \cdot \|$, therefore,

$$\mathcal{N}(\mathcal{F}_\mathrm{OE}, M_{\ell_\mathrm{OE}} \epsilon, L^\infty) \le \mathcal{N}(\mathcal{W}, M_{\ell_\mathrm{OE}} \epsilon / L_\mathrm{OE}, \| \cdot \|) \le (1 + \frac{\mathrm{diam}(\mathcal{W}) L_\mathrm{OE}}{M_{\ell_\mathrm{OE}} \epsilon})^{d'},$$

which implies that

$$\int_0^1 \sqrt{\log(\mathcal{N}(\mathcal{F}_{\mathrm{OE}}, M_{\ell_{\mathrm{OE}}}\epsilon, L^\infty))}\mathrm{d}\epsilon$$

$$\leq \sqrt{d'} \int_0^1 \sqrt{\log(1 + \frac{\mathrm{diam}(\mathcal{W})L_{\mathrm{OE}}}{M_{\ell_{\mathrm{OE}}}\epsilon})}\mathrm{d}\epsilon$$

$$\leq \sqrt{d'} \int_0^1 \sqrt{\frac{\mathrm{diam}(\mathcal{W})L_{\mathrm{OE}}}{M_{\ell_{\mathrm{OE}}}\epsilon}}\mathrm{d}\epsilon = 2\sqrt{\frac{\mathrm{diam}(\mathcal{W})L_{\mathrm{OE}}d'}{M_{\ell_{\mathrm{OE}}}}}.$$

By Lemma 4, we obtain that there exist two uniform constants such that with the probability at least $1 - e^{-t} > 0$,

$$\sup_{\mathrm{W}_c(D_{X'},D_{X_{\mathrm{A}}})\leq\rho} \mathbb{E}_{\mathbf{x}\sim D_{\mathrm{O}}}\ell_{\mathrm{OE}}(\mathbf{f_w};\mathbf{x})$$

$$\leq \sup_{\mathrm{W}_c(D_{X'},\widehat{D}_{X_{\mathrm{A}}})\leq\rho} \mathbb{E}_{\mathbf{x}\sim D_{\mathrm{O}}}\ell_{\mathrm{OE}}(\mathbf{f_w};\mathbf{x}) + b_2 M_{\ell_{\mathrm{OE}}}\sqrt{\frac{t}{m}} + b_1 M_{\ell_{\mathrm{OE}}}\sqrt{\frac{\mathrm{diam}(\mathcal{W})L_{\mathrm{OE}}d'}{\rho^2 m}},$$

where $b_1$ and $b_2$ are uniform constants. $\qquad\square$

**Lemma 6.** *Given the same assumptions in Theorem 2, and for any* $\mathbf{w} \in \mathcal{W}$ *and any* $\mathbf{x}, \mathbf{x}' \in \mathcal{X}$,

$$|\ell_{\mathrm{OE}}(\mathbf{f_w};\mathbf{x}) - \ell_{\mathrm{OE}}(\mathbf{f_w};\mathbf{x}')| \leq L_c c(\mathbf{x}, \mathbf{x}'),$$

*then for any* $\delta \geq 0$,

$$\sup_{\mathrm{W}_c(D_{X'},D_{X_{\mathrm{A}}})\leq\rho+\delta} \mathbb{E}_{\mathbf{x}\sim D_{X'}}\ell_{\mathrm{OE}}(\mathbf{f_w};\mathbf{x}) - \sup_{\mathrm{W}_c(D_{X'},D_{X_{\mathrm{A}}})\leq\rho} \mathbb{E}_{\mathbf{x}\sim D_{X'}}\ell_{\mathrm{OE}}(\mathbf{f_w};\mathbf{x}) \leq L_c\delta.$$

*Proof of Lemma 6.* For each $\epsilon > 0$, we set $D_{X'}^{\delta,\epsilon}$ satisfies that

$$\sup_{\mathrm{W}_c(D_{X'},D_{X_{\mathrm{A}}})\leq\rho+\delta} \mathbb{E}_{\mathbf{x}\sim D_{X'}}\ell_{\mathrm{OE}}(\mathbf{f_w};\mathbf{x}) \leq \mathbb{E}_{\mathbf{x}\sim D_{X'}^{\delta,\epsilon}}\ell_{\mathrm{OE}}(\mathbf{f_w};\mathbf{x}) + \epsilon,$$

and

$$\mathrm{W}_c(D_{X'}^{\delta,\epsilon}, D_{X_{\mathrm{A}}}) \leq \rho + \delta.$$

**Case 1.** If

$$\mathrm{W}_c(D_{X'}^{\delta,\epsilon}, D_{X_{\mathrm{A}}}) \leq \rho,$$

then

$$\sup_{\mathrm{W}_c(D_{X'},D_{X_{\mathrm{A}}})\leq\rho+\delta} \mathbb{E}_{\mathbf{x}\sim D_{X'}}\ell_{\mathrm{OE}}(\mathbf{f_w};\mathbf{x}) - \sup_{\mathrm{W}_c(D_{X'},D_{X_{\mathrm{A}}})\leq\rho} \mathbb{E}_{\mathbf{x}\sim D_{X'}}\ell_{\mathrm{OE}}(\mathbf{f_w};\mathbf{x}) \leq \epsilon.$$

**Case 2.** If

$$\mathrm{W}_c(D_{X'}^{\delta,\epsilon}, D_{X_{\mathrm{A}}}) > \rho,$$

then we consider a special distribution $D'_{X'}$, which is defined as follows: for some $u \in [0, 1]$,

$$D'_{X'} = (1 - u)D_{X'}^{\delta,\epsilon} + uD_{X_{\mathrm{A}}}.$$

It is clear that

$$\mathrm{W}_c(D'_{X'}, D_{X_{\mathrm{A}}}) \leq (1 - u)\mathrm{W}_c(D_{X'}^{\delta,\epsilon}, D_{X_{\mathrm{A}}}) \leq (1 - u)(\rho + \delta).$$

hence, if we set $u = \delta/(\rho + \delta)$,

$$\mathrm{W}_c(D'_{X'}, D_{X_{\mathrm{A}}}) \leq \rho.$$

Because $c(\cdot, \cdot)$ is a metric, Kantorovich–Rubinstein duality Villani (2021) implies that

$$\mathrm{W}_c(D'_{X'}, D_{X'}^{\delta,\epsilon})$$

$$= \sup_{\|f\|_{\mathrm{Lip}}\leq 1} \int_{\mathcal{X}} f(\mathbf{x})\mathrm{d}D'_{X'}(\mathbf{x}) - \int_{\mathcal{X}} f(\mathbf{x})\mathrm{d}D_{X'}^{\delta,\epsilon}(\mathbf{x})$$

$$= u \sup_{\|f\|_{\mathrm{Lip}}\leq 1} \int_{\mathcal{X}} f(\mathbf{x})\mathrm{d}D_{X_{\mathrm{A}}}(\mathbf{x}) - \int_{\mathcal{X}} f(\mathbf{x})\mathrm{d}D_{X'}^{\delta,\epsilon}(\mathbf{x})$$

$$= u\mathrm{W}_c(D_{X_{\mathrm{A}}}, D_{X'}^{\delta,\epsilon})$$

$$= \delta$$

By Kantorovich–Rubinstein duality (Villani, 2021), we also obtain that

$$\sup_{W_c(D_{X'}, D_{X_A}) \leq \rho + \delta} \mathbb{E}_{\mathbf{x} \sim D_{X'}} \ell_{\mathrm{OE}}(\mathbf{f_w}; \mathbf{x}) - \sup_{W_c(D_{X'}, D_{X_A}) \leq \rho} \mathbb{E}_{\mathbf{x} \sim D_{X'}} \ell_{\mathrm{OE}}(\mathbf{f_w}; \mathbf{x})$$

$$\leq \mathbb{E}_{\mathbf{x} \sim D_{X'}^{\delta, \epsilon}} \ell_{\mathrm{OE}}(\mathbf{f_w}; \mathbf{x}) - \mathbb{E}_{\mathbf{x} \sim D_{X'}'} \ell_{\mathrm{OE}}(\mathbf{f_w}; \mathbf{x}) + \epsilon \leq L_c \delta + \epsilon,$$

which implies that

$$\sup_{W_c(D_{X'}, D_{X_A}) \leq \rho + \delta} \mathbb{E}_{\mathbf{x} \sim D_{X'}} \ell_{\mathrm{OE}}(\mathbf{f_w}; \mathbf{x}) - \sup_{W_c(D_{X'}, D_{X_A}) \leq \rho} \mathbb{E}_{\mathbf{x} \sim D_{X'}} \ell_{\mathrm{OE}}(\mathbf{f_w}; \mathbf{x}) \leq L_c \delta.$$

By Cases 1 and 2, we prove this lemma. □

**Lemma 7.** *Given the same assumptions in Theorem 2, if*

- *$\ell_{\mathrm{OE}}(\cdot; \mathbf{x})$ is $L_{\mathrm{OE}}$-Lipschitz w.r.t. norm $\| \cdot \|$ for all $\mathbf{x} \in \mathcal{X}$;*

- *the parameter space $\mathcal{W} \subset \mathbb{R}^{d'}$ satisfies that*

$$\mathrm{diam}(\mathcal{W}) = \sup_{\mathbf{w}, \mathbf{w}' \in \mathcal{W}} \|\mathbf{w} - \mathbf{w}'\| < +\infty;$$

- *for each $\mathbf{w} \in \mathcal{W}$ and any $\mathbf{x}, \mathbf{x}' \in \mathcal{X}$,*

$$|\ell_{\mathrm{OE}}(\mathbf{f_w}; \mathbf{x}) - \ell_{\mathrm{OE}}(\mathbf{f_w}; \mathbf{x}')| \leq L_c c(\mathbf{x}, \mathbf{x}'),$$

*then with the probability at least $1 - e^{-t} > 0$, for any $\rho > 0$ and $\mathbf{w} \in \mathcal{W}$,*

$$\sup_{W_c(D_{X'}, D_{X_A}) \leq \rho} \mathbb{E}_{\mathbf{x} \sim D_{X'}} \ell_{\mathrm{OE}}(\mathbf{f_w}; \mathbf{x})$$

$$\leq \sup_{W_c(D_{X'}, \widehat{D}_{X_A}) \leq \rho} \mathbb{E}_{\mathbf{x} \sim D_{X'}} \ell_{\mathrm{OE}}(\mathbf{f_w}; \mathbf{x}) + b_2 M_{\ell_{\mathrm{OE}}} \sqrt{\frac{t}{m}} + b_1 \min\{L_c, \frac{M_{\ell_{\mathrm{OE}}}}{\rho}\} \sqrt{\frac{\mathrm{diam}(\mathcal{W}) L_{\mathrm{OE}} d'}{m}},$$

(17)

*where $b_1$ and $b_2$ are uniform constants.*

*Proof.* By Lemma 3 and the similar proving process in Lemma 5, we obtain that with the probability at least $1 - e^{-t} > 0$, for any $\gamma \geq 0$, $\rho \geq 0$ and $\mathbf{w} \in \mathcal{W}$, we have

$$\sup_{W_c(D_{X'}, D_{X_A}) \leq \rho} \mathbb{E}_{\mathbf{x} \sim D_{X'}} \ell_{\mathrm{OE}}(\mathbf{f_w}; \mathbf{x})$$

$$\leq \gamma \rho + \frac{1}{m} \sum_{i=1}^{m} \phi_\gamma(\mathbf{w}; \mathbf{x}_A^i) + b_2 M_{\ell_{\mathrm{OE}}} \sqrt{\frac{t}{m}} + \gamma b_1 \sqrt{\frac{\mathrm{diam}(\mathcal{W}) L_{\mathrm{OE}} d'}{m}},$$

(18)

where $b_1$ and $b_2$ are uniform constants.

Let

$$\rho_m = \rho + b_1 \sqrt{\frac{\mathrm{diam}(\mathcal{W}) L_{\mathrm{OE}} d'}{m}},$$

and

$$\Delta_m = \sup_{W_c(D_{X'}, \widehat{D}_{X_A}) \leq \rho_m} \mathbb{E}_{\mathbf{x} \sim D_{X'}} \ell_{\mathrm{OE}}(\mathbf{f_w}; \mathbf{x}) - \sup_{W_c(D_{X'}, \widehat{D}_{X_A}) \leq \rho} \mathbb{E}_{\mathbf{x} \sim D_{X'}} \ell_{\mathrm{OE}}(\mathbf{f_w}; \mathbf{x}).$$

Note that

$$\inf_{\gamma \geq 0} \gamma \rho_m + \frac{1}{m} \sum_{i=1}^{m} \phi_\gamma(\mathbf{w}; \mathbf{x}_A^i) = \sup_{W_c(D_{X'}, \widehat{D}_{X_A}) \leq \rho_m} \mathbb{E}_{\mathbf{x} \sim D_{X'}} \ell_{\mathrm{OE}}(\mathbf{f_w}; \mathbf{x}),$$

and by Lemma 6,

$$\Delta_m \leq L_c(\rho_m - \rho) = b_1 L_c \sqrt{\frac{\mathrm{diam}(\mathcal{W}) L_{\mathrm{OE}} d'}{m}},$$

hence, by Eq. (18), we know that with the probability at least $1 - e^{-t} > 0$, for any $\rho \geq 0$ and $\mathbf{w} \in \mathcal{W}$,

$$\sup_{\mathrm{W}_c(D_{X'}, D_{X_A}) \leq \rho} \mathbb{E}_{\mathbf{x} \sim D_{X'}} \ell_{\mathrm{OE}}(\mathbf{f_w}; \mathbf{x})$$

$$\leq \sup_{\mathrm{W}_c(D_{X'}, \widehat{D}_{X_A}) \leq \rho} \mathbb{E}_{\mathbf{x} \sim D_{X'}} \ell_{\mathrm{OE}}(\mathbf{f_w}; \mathbf{x}) + b_1 L_c \sqrt{\frac{\mathrm{diam}(\mathcal{W}) L_{\mathrm{OE}} d'}{m}} + b_2 M_{\ell_{\mathrm{OE}}} \sqrt{\frac{t}{m}}, \quad (19)$$

where $b_1$ and $b_2$ are uniform constants.

Combining Lemma 3 with Eq. (19), we know that with the probability at least $1 - e^{-t} > 0$, for any $\rho > 0$ and $\mathbf{w} \in \mathcal{W}$,

$$\sup_{\mathrm{W}_c(D_{X'}, D_{X_A}) \leq \rho} \mathbb{E}_{\mathbf{x} \sim D_{X'}} \ell_{\mathrm{OE}}(\mathbf{f_w}; \mathbf{x})$$

$$\leq \sup_{\mathrm{W}_c(D_{X'}, \widehat{D}_{X_A}) \leq \rho} \mathbb{E}_{\mathbf{x} \sim D_{X'}} \ell_{\mathrm{OE}}(\mathbf{f_w}; \mathbf{x}) + b_2 M_{\ell_{\mathrm{OE}}} \sqrt{\frac{t}{m}} + b_1 \min\{L_c, \frac{M_{\ell_{\mathrm{OE}}}}{\rho}\} \sqrt{\frac{\mathrm{diam}(\mathcal{W}) L_{\mathrm{OE}} d'}{m}},$$

where $b_1$ and $b_2$ are uniform constants. $\qquad \square$

**Lemma 8.** *Given the same assumptions in Lemma 4, for a fixed $\mathbf{w}_0 \in \mathcal{W}$, then with the probability at least $1 - e^{-t} > 0$, for any $\rho \geq 0$*

$$\sup_{\mathrm{W}_c(D_{X'}, \widehat{D}_{X_A}) \leq \rho} \mathbb{E}_{\mathbf{x} \sim D_{X'}} \ell_{\mathrm{OE}}(\mathbf{f}_{\mathbf{w}_0}; \mathbf{x}) \leq \sup_{\mathrm{W}_c(D_{X'}, D_{X_A}) \leq \rho} \mathbb{E}_{\mathbf{x} \sim D_{X'}} \ell_{\mathrm{OE}}(\mathbf{f}_{\mathbf{w}_0}; \mathbf{x}) + 2 M_{\ell_{\mathrm{OE}}} \sqrt{\frac{2t}{m}}.$$

*Proof of Lemma 8.* By Sinha et al. (2018), it is clear that

$$\sup_{\mathrm{W}_c(D_{X'}, D_{X_A}) \leq \rho} \mathbb{E}_{\mathbf{x} \sim D_{X'}} \ell_{\mathrm{OE}}(\mathbf{f}_{\mathbf{w}_0}; \mathbf{x}) = \inf_{\gamma \geq 0} [\gamma \rho + \mathbb{E}_{\mathbf{x} \sim D_{X_A}} \phi_\gamma(\mathbf{w}_0; \mathbf{x})]$$

Therefore, for each $\epsilon > 0$, there exists a constant $\gamma_\epsilon \geq 0$ such that

$$\gamma_\epsilon \rho + \mathbb{E}_{\mathbf{x} \sim D_{X_A}} \phi_{\gamma_\epsilon}(\mathbf{w}_0; \mathbf{x}) \leq \sup_{\mathrm{W}_c(D_{X'}, D_{X_A}) \leq \rho} \mathbb{E}_{\mathbf{x} \sim D_{X'}} \ell_{\mathrm{OE}}(\mathbf{f}_{\mathbf{w}_0}; \mathbf{x}) + \epsilon.$$

Combining the above inequality, Lemma 2 and McDiarmid's Inequality, then with the probability at least

$$1 - \exp\left(\frac{-\epsilon_0^2 m}{2 M_{\ell_{\mathrm{OE}}}^2}\right) > 0,$$

we have

$$\mathbb{E}_{\mathbf{x} \sim \widehat{D}_{X_A}} \phi_{\gamma_\epsilon}(\mathbf{w}_0; \mathbf{x}) \leq \mathbb{E}_{\mathbf{x} \sim D_{X_A}} \phi_{\gamma_\epsilon}(\mathbf{w}_0; \mathbf{x}) + \epsilon_0.$$

If we set $t = \epsilon_0^2 m / 2 M_{\ell_{\mathrm{OE}}}^2$, then

$$\epsilon_0 = M_{\ell_{\mathrm{OE}}} \sqrt{\frac{2t}{m}}.$$

Hence, with the probability at least $1 - e^{-t} > 0$, we have

$$\gamma_\epsilon \rho + \mathbb{E}_{\mathbf{x} \sim \widehat{D}_{X_A}} \phi_{\gamma_\epsilon}(\mathbf{w}_0; \mathbf{x}) \leq \sup_{\mathrm{W}_c(D_{X'}, D_{X_A}) \leq \rho} \mathbb{E}_{\mathbf{x} \sim D_{X'}} \ell_{\mathrm{OE}}(\mathbf{f}_{\mathbf{w}_0}; \mathbf{x}) + \epsilon + M_{\ell_{\mathrm{OE}}} \sqrt{\frac{2t}{m}},$$

which implies that with the probability at least $1 - e^{-t} > 0$,

$$\sup_{\mathrm{W}_c(D_{X'}, \widehat{D}_{X_A}) \leq \rho} \mathbb{E}_{\mathbf{x} \sim D_{X'}} \ell_{\mathrm{OE}}(\mathbf{f}_{\mathbf{w}_0}; \mathbf{x}) \leq \sup_{\mathrm{W}_c(D_{X'}, D_{X_A}) \leq \rho} \mathbb{E}_{\mathbf{x} \sim D_{X'}} \ell_{\mathrm{OE}}(\mathbf{f}_{\mathbf{w}_0}; \mathbf{x}) + \epsilon + M_{\ell_{\mathrm{OE}}} \sqrt{\frac{2t}{m}},$$

because

$$\gamma_\epsilon \rho + \mathbb{E}_{\mathbf{x} \sim \widehat{D}_{X_A}} \phi_{\gamma_\epsilon}(\mathbf{w}_0; \mathbf{x}) \geq \sup_{\mathrm{W}_c(D_{X'}, \widehat{D}_{X_A}) \leq \rho} \mathbb{E}_{\mathbf{x} \sim D_{X'}} \ell_{\mathrm{OE}}(\mathbf{f}_{\mathbf{w}_0}; \mathbf{x}).$$

By setting $\epsilon = M_{\ell_{\mathrm{OE}}} \sqrt{2t/m}$, we complete this proof. $\qquad \square$

**Lemma 9.** *If $0 \leq \ell(\mathbf{f_w}; \mathbf{x}, y) \leq M_\ell$, then with the probability at least $1 - e^{-t} > 0$, we have that for any $\mathbf{w} \in \mathcal{W}$,*

$$\mathbb{E}_{(\mathbf{x}, y) \sim D_{X_I Y_I}} \ell(\mathbf{f_w}; \mathbf{x}, y) - \frac{1}{n} \sum_{i=1}^{n} \ell(\mathbf{f_w}; \mathbf{x}_I^i, y_I^i) \leq \frac{b_0 M_\ell}{\sqrt{n}} \int_0^1 \sqrt{\log \mathcal{N}(\mathcal{F}, M_\ell \epsilon, L^\infty)} d\epsilon + M_\ell \sqrt{\frac{2t}{n}},$$

*where $b_0$ is a uniform constant.*

*Proof of Lemma 9.* Let

$$X_{\ell(\mathbf{f_w};\cdot)} = \mathbb{E}_{(\mathbf{x}, y) \sim D_{X_I Y_I}} \ell(\mathbf{f_w}; \mathbf{x}, y) - \frac{1}{n} \sum_{i=1}^{n} \ell(\mathbf{f_w}; \mathbf{x}_I^i, y_I^i).$$

Then, it is clear that

$$\mathbb{E}_{S \sim D_{X_I Y_I}^n} X_{\ell(\mathbf{f_w};\cdot)} = 0.$$

By Proposition 2.6.1 and Lemma 2.6.8 in Vershynin (2018),

$$\|X_{\ell(\mathbf{f_w};\cdot)} - X_{\ell(\mathbf{f_{w'}};\cdot)}\|_{\Phi_2} \leq \frac{c_0}{\sqrt{n}} \|\ell(\mathbf{f_w}; \cdot) - \ell(\mathbf{f_{w'}}; \cdot)\|_{L^\infty},$$

where $\| \cdot \|_{\Phi_2}$ is the sub-gaussian norm and $c_0$ is a uniform constant. Therefore, by Dudley's entropy integral (Vershynin, 2018), we have

$$\mathbb{E}_{S \sim D_{X_I Y_I}^n} \sup_{\mathbf{w} \in \mathcal{W}} X_{\ell(\mathbf{f_w};\cdot)} \leq \frac{b_0}{\sqrt{n}} \int_0^{+\infty} \sqrt{\log \mathcal{N}(\mathcal{F}, \epsilon, L^\infty)} d\epsilon,$$

where $b_0$ is a uniform constant and

$$\mathcal{F} = \{\ell(\mathbf{f_w}; \mathbf{x}, y) : \mathbf{w} \in \mathcal{W}\}.$$

Note that

$$\begin{aligned}
\mathbb{E}_{S \sim D_{X_I Y_I}^n} \sup_{\mathbf{w} \in \mathcal{W}} X_{\ell(\mathbf{f_w};\cdot)} &\leq \frac{b_0}{\sqrt{n}} \int_0^{+\infty} \sqrt{\log \mathcal{N}(\mathcal{F}, \epsilon, L^\infty)} d\epsilon \\
&= \frac{b_0}{\sqrt{n}} \int_0^{M_\ell} \sqrt{\log \mathcal{N}(\mathcal{F}, \epsilon, L^\infty)} d\epsilon \\
&= \frac{b_0}{\sqrt{n}} M_\ell \int_0^1 \sqrt{\log \mathcal{N}(\mathcal{F}, M_\ell \epsilon, L^\infty)} d\epsilon.
\end{aligned}$$

Then, similar to the proof of Lemma 8, we use McDiarmid's Inequality, then with the probability at least $1 - e^{-t} > 0$, for any $\mathbf{w} \in \mathcal{W}$,

$$X_{\ell(\mathbf{f_w};\cdot)} \leq \frac{b_0}{\sqrt{n}} M_\ell \int_0^1 \sqrt{\log \mathcal{N}(\mathcal{F}, M_\ell \epsilon, L^\infty)} d\epsilon + M_\ell \sqrt{\frac{2t}{n}}.$$

$\square$

**Lemma 10.** *If $0 \leq \ell(\mathbf{f_w}; \mathbf{x}, y) \leq M_\ell$, then for a fixed $\mathbf{w}_0 \in \mathcal{W}$, with the probability at least $1 - e^{-t} > 0$,*

$$\frac{1}{n} \sum_{i=1}^{n} \ell(\mathbf{f_{w_0}}; \mathbf{x}_I^i, y_I^i) - \mathbb{E}_{(\mathbf{x}, y) \sim D_{X_I Y_I}} \ell(\mathbf{f_{w_0}}; \mathbf{x}, y) \leq M_\ell \sqrt{\frac{2t}{n}}.$$

*Proof of Lemma 10.* Similar to the proof of Lemma 8, McDiarmid's Inequality implies this result.
$\square$

**Lemma 11.** *If*

- $0 \leq \ell(\mathbf{f_w}; \mathbf{x}, y) \leq M_\ell$,
- $\ell(\mathbf{f_w}; \mathbf{x}, y)$ *is $L$-Lipschitz w.r.t. norm $\| \cdot \|$, i.e., for any $(\mathbf{x}, y) \in \mathcal{X} \times \mathcal{Y}$, and $\mathbf{w}, \mathbf{w'} \in \mathcal{W}$,*

$$|\ell(\mathbf{f_w}; \mathbf{x}, y) - \ell(\mathbf{f_{w'}}; \mathbf{x}, y)| \leq L \|\mathbf{w} - \mathbf{w'}\|,$$

- *the parameter space $\mathcal{W} \subset \mathbb{R}^{d'}$ satisfies that*

$$\mathrm{diam}(\mathcal{W}) = \sup_{\mathbf{w}, \mathbf{w}' \in \mathcal{W}} \|\mathbf{w} - \mathbf{w}'\| < +\infty,$$

*then with the probability at least $1 - e^{-t} > 0$, we have that for any $\mathbf{w} \in \mathcal{W}$,*

$$\mathbb{E}_{(\mathbf{x}, y) \sim D_{X_\mathrm{I} Y_\mathrm{I}}} \ell(\mathbf{w}; \mathbf{x}, y) - \frac{1}{n} \sum_{i=1}^{n} \ell(\mathbf{w}; \mathbf{x}_\mathrm{I}^i, y_\mathrm{I}^i) \leq b_0 \sqrt{\frac{M_\ell \mathrm{diam}(\mathcal{W}) L d'}{n}} + M_\ell \sqrt{\frac{2t}{n}},$$

*where $b_0$ is a uniform constant.*

*Proof of Lemma 11.* The proof is similar to Corollary 1 in Sinha et al. (2018) and Lemma 5. Note that

$$\mathcal{F} = \{\ell(\mathbf{f_w}; \mathbf{x}, y) : \mathbf{w} \in \mathcal{W}\},$$

and $\ell(\mathbf{f_w}; \mathbf{x}, y)$ is $L$-Lipschitz w.r.t. norm $\|\cdot\|$, therefore,

$$\mathcal{N}(\mathcal{F}, M_\ell \epsilon, L^\infty) \leq \mathcal{N}(\mathcal{W}, M_\ell \epsilon / L, \|\cdot\|) \leq (1 + \frac{\mathrm{diam}(\mathcal{W}) L}{M_\ell \epsilon})^{d'},$$

which implies that

$$\int_0^1 \sqrt{\log(\mathcal{N}(\mathcal{F}, M_\ell \epsilon, L^\infty)} \mathrm{d}\epsilon \leq \sqrt{d'} \int_0^1 \sqrt{\log(1 + \frac{\mathrm{diam}(\mathcal{W}) L}{M_\ell \epsilon})} \mathrm{d}\epsilon$$

$$\leq \sqrt{d'} \int_0^1 \sqrt{\frac{\mathrm{diam}(\mathcal{W}) L}{M_\ell \epsilon}} \mathrm{d}\epsilon = 2 \sqrt{\frac{\mathrm{diam}(\mathcal{W}) L d'}{M_\ell}}.$$

By Lemma 9, we obtain this result. $\qquad\square$

# E  Further Discussions

As discussed in Section 4.2, we realize the dual optimization objective following Eq. (9), searching for the worst OOD data in a finite-dimensional space to ease the computation. Furthermore, directly searching in the input space is typically hard for optimization (Madry et al., 2018; Wang et al., 2021b), where the results can easily stuck at sub-optimal solutions and the computation is time-consuming. Therefore, we suggest perturbing the worst OOD data in the embedding space. Denote the embedding features of an input by $\mathbf{e}(\mathbf{x})$, we consider the bi-level optimization problem:

$$\inf_{\gamma \geq 0} \left\{ \gamma\rho + \frac{1}{m} \sum_{i=1}^{m} \phi_\gamma(\mathbf{w}; \mathbf{e}(\mathbf{x}_A^i)) \right\},$$

$$\text{where } \phi_\gamma(\mathbf{w}; \mathbf{e}(\mathbf{x}_A^i)) = \sup_{\mathbf{p}^i \in \mathcal{E}} \left\{ \ell_{\text{OE}}(\mathbf{h}(\mathbf{e}(\mathbf{x}_A^i) + \mathbf{p}^i); \mathbf{e}(\mathbf{x}_A^i)) - \gamma\|\mathbf{p}^i\|_1 \right\}.$$

Such an bi-level problem can be solved by alternative optimization (Huang et al., 2023b; Liu et al., 2021): we first find the proper value of $\mathbf{p}^i$ that approaches to the true value of $\phi_\gamma(\mathbf{w}; \mathbf{e}(\mathbf{x}_A^i))$, and then we update the value of $\gamma$ that leads to the smallest value of $\gamma\rho + \frac{1}{m} \sum_{i=1}^{m} \phi_\gamma(\mathbf{w}; \mathbf{e}(\mathbf{x}_A^i))$. The gradient descent/ascent can be adopted for optimization. Specifically, for the perturbation $\mathbf{p}^i$, each optimization step is

$$\phi_\gamma(\mathbf{w}; \mathbf{e}(\mathbf{x}_A^i)) \leftarrow \ell_{\text{OE}}\left(\mathbf{h}(\mathbf{e}(\mathbf{x}_A^i) + \mathbf{p}^i); \mathbf{e}(\mathbf{x}_A^i)\right) - \gamma\left\|\mathbf{p}^i\right\|_1,$$

$$\mathbf{p}^i \leftarrow \mathbf{p}^i + \text{ps}\nabla_{\mathbf{p}^i}\phi_\gamma(\mathbf{w}; \mathbf{e}(\mathbf{x}_A^i)),$$

where ps is the learning rate. For $\gamma$, the optimization step follows:

$$\gamma \leftarrow \gamma - \beta\left\{\rho - \frac{1}{m} \sum_{i=1}^{m} \left\|\mathbf{p}^i\right\|_1\right\},$$

with $\beta$ the learning rate. Furthermore, to avoid the extreme value and/or the negative value of $\gamma$, we should conduct value clipping for $\gamma$, which is given by

$$\gamma \leftarrow \min(\max(\gamma, \gamma_{\max}), 0),$$

where we constrain the minimum of $\gamma$ to be 0 and the maximum of $\gamma$ to be $\gamma_{\max}$.

## E.1  Understanding Theoretical Results

Theorem 2 justifies that when the model complexity and the sample size are large enough, the empirical solution given by our DAL risk will converge to its optimal value, i.e., $\min_{\mathbf{w}} R_D(\mathbf{w}; \rho)$. Therefore, the difference between the expected and the empirical error is bounded w.r.t. the DAL risk. Theorem 3 goes one step further, studying the detection risk w.r.t. (unseen) real OOD data. It states that the open-world performance of our DAL depends on both the approximate risk and the estimation error. The former models the best performance (i.e., Bayes optimal) that our DAL can achieve, and the latter depends on the OOD distribution gap, the radius, and the excess error introduced in Theorem 2. In summary, Theorem 2 considers the convergence for DAL itself, while Theorem 3 justifies that DAL can mitigate the OOD distribution discrepancy in the open world.

## E.2  Comparison with DOE

A parallel work, named DOE (Wang et al., 2023), also focuses on mitigating the OOD distribution discrepancy issue. Overall, they state that model perturbation can lead to input transformation, and thus learning from the perturbed model can make the predictor learn from diverse distributions with respect to auxiliary OOD cases. Moreover, to make the transformed data benefit the model most, DOE searches for the associated perturbation that leads to the worst OOD regret.

Similar to DOE, we also learn from the worst OOD cases to mitigate the distribution discrepancy, but DAL further enjoys the following two strengths. 1) From the theoretical perspective, our clear definition of the candidate OOD distribution space, i.e., the Wasserstein ball, allows us to investigate the impact of DAL for open-world OOD detection (cf., Theorem 3). In contrast, DOE only constrains the magnitude of the perturbation strength, making it limited to proving convergence w.r.t. their

Table 4: ID accuracy on the CIFAR benchmarks for those methods that require model training.

| Method | ERM | CSI | VOS | OE | Energy-OE | ATOM | DOE | POEM | DAL |
|--------|------|------|------|------|-----------|------|------|------|------|
| CIFAR-10 | 94.28 | 94.33 | 94.58 | 95.22 | 94.84 | 95.12 | 94.28 | 93.32 | 95.01 |
| CIFAR-100 | 73.98 | 74.30 | 75.50 | 75.90 | 71.61 | 74.04 | 74.51 | 74.85 | 76.13 |

proposed learning objective (cf., Theorem 2 in Wang et al. (2023)). 2) From the algorithmic perspective, we directly search for the worst OOD data in the embedding space, more effective than DAL, which requires searching the model perturbation for the whole model. As a result, our theoretically-driven framework, i,e, DAL, yields superior performance over DOE in Table 1 while requiring less computation cost (DAL take only half the training time compared with DOE per epoch).

### E.3 Discussing about Limitations

In theory, our main drawback lies in the trade-off between estimation and approximation errors (cf., Theorem 3), where we may not get a very tight bound for the real OOD risk. In algorithm, the worst OOD data are constrained in the ball around the auxiliary OOD data (cf., Algorithm 1), of which the coverage may not include real OOD data. Moreover, we conduct distribution augmentation in the embedding space, where our Theorem 3 can only be applied. Other data generation approaches, which can lead to more complex forms of distribution augmentation in the input space, are also of interest. Our future studies will focus on advanced learning schemes that address the above issues, e.g., modeling the data generation process through the causality perspective (Zhang et al., 2022).

## F Further Evaluations

We provide more information about evaluation setups and conduct additional experiments on DAL.

### F.1 Hardware Configurations

All experiments are realized by Pytorch 1.81 with CUDA 11.1, using machines equipped with GeForce RTX 3090 GPUs and AMD Threadripper 3960X Processors.

### F.2 ID Accuracy

We report the ID accuracy for those methods that require model training on the CIFAR benchmarks, of which the results are summarized in Table 4. We also list the results for the model conventionally trained on ID data with empirical risk minimization (ERM). Overall, most of the considered methods can preserve relatively high ID accuracy. Moreover, those methods that regularize predictors by auxiliary OOD data, such as OE and DAL, can even show further improvements. It indicates that learning with auxiliary data can achieve high detection performance and maintain good ID accuracy.

### F.3 Other Scoring Functions

We further claim that many advanced scoring strategies other than MSP can also benefit from DAL. In Table 5, we compare the OOD detection performance before (w/o train) and after (w/ DAL) DAL training across a set of representative scoring strategies, including MSP, Fee Energy, ASH, Mahalanobis, and KNN. We also compare the results after OE training (w/ OE). As we can see, both OE and DAL can lead to much better results than before training, and DAL can further boost detection performance over OE. It indicates that the benefits of our proposal are not limited to the specific scoring function of MSP. However, Mahalanobis fails (FPR95 more than 95) on CIFAR-100 after OE and DAL training, which may require further exploration.

### F.4 Mean and Standard Deviation

This section validates the experiments on CIFAR benchmarks with five individual trials (random seeds), comparing between our DAL and OE. Besides the individual results, we also summarize the mean performance and standard deviation across the trails for both CIFAR-10 and CIFAR-100. We

Table 5: Comparison on the CIFAR benchmarks with different scoring strategies.

| | MSP | | Free Energy | | ASH | | Mahalanobis | | KNN | |
|---|---|---|---|---|---|---|---|---|---|---|
| | FPR | AUROC | FPR | AUROC | FPR | AUROC | FPR | AUROC | FPR | AUROC |
| | | | | | CIFAR-10 | | | | | |
| w/o train | 50.15 | 91.02 | 33.21 | 91.01 | 32.98 | 91.85 | 46.64 | 88.59 | 33.38 | 93.76 |
| w/ OE | 4.67 | 98.88 | 3.40 | 98.98 | 3.35 | 98.99 | 15.80 | 94.32 | 5.50 | 98.71 |
| w/ DAL | 2.68 | 99.01 | 2.59 | 98.99 | 2.50 | 98.70 | 12.75 | 95.55 | 5.04 | 97.58 |
| | | | | | CIFAR-100 | | | | | |
| w/o train | 78.61 | 75.95 | 69.84 | 75.20 | 59.31 | 84.46 | 72.37 | 82.70 | 59.31 | 84.46 |
| w/ OE | 43.14 | 90.27 | 36.98 | 92.66 | 33.82 | 93.36 | - | - | 53.14 | 83.50 |
| w/ DAL | 29.68 | 93.92 | 29.63 | 93.83 | 29.73 | 94.05 | - | - | 50.46 | 84.75 |

Table 6: Comparison of DAL and outlier exposure on CIFAR-10 with 5 individual trails. $\downarrow$ (or $\uparrow$) indicates smaller (or larger) values are preferred; and a bold font indicates the best results in the corresponding column.

| Trails | SVHN | | LSUN | | iSUN | | Textures | | Places365 | | Average | |
|---|---|---|---|---|---|---|---|---|---|---|---|---|
| | FPR95 $\downarrow$ | AUROC $\uparrow$ | FPR95 $\downarrow$ | AUROC $\uparrow$ | FPR95 $\downarrow$ | AUROC $\uparrow$ | FPR95 $\downarrow$ | AUROC $\uparrow$ | FPR95 $\downarrow$ | AUROC $\uparrow$ | FPR95 $\downarrow$ | AUROC $\uparrow$ |
| | | | | | | OE | | | | | | |
| #1 | 1.50 | 99.23 | 1.10 | 99.33 | 1.70 | 99.18 | 4.00 | 98.64 | 11.30 | 97.09 | 3.92 | 98.69 |
| #2 | 1.25 | 99.15 | 1.05 | 99.49 | 2.20 | 98.88 | 4.15 | 98.59 | 11.60 | 97.08 | 4.05 | 98.63 |
| #3 | 1.25 | 99.38 | 1.05 | 99.42 | 1.75 | 99.01 | 4.00 | 98.82 | 11.10 | 97.04 | 3.83 | 98.73 |
| #4 | 1.70 | 99.13 | 1.05 | 99.52 | 2.10 | 99.12 | 4.20 | 98.55 | 11.65 | 97.08 | 4.14 | 98.68 |
| #5 | 1.35 | 99.17 | 1.30 | 99.49 | 1.40 | 99.26 | 4.60 | 99.00 | 11.75 | 97.03 | 1.08 | 98.79 |
| mean | 1.41 | 99.21 | 1.10 | 99.45 | 1.83 | 99.09 | 4.19 | 98.72 | 11.48 | 97.06 | 3.40 | 98.70 |
| $\pm$ std | $\pm$ 0.17 | $\pm$ 0.10 | $\pm$ 0.02 | $\pm$ 0.07 | $\pm$ 0.28 | $\pm$ 0.15 | $\pm$ 0.22 | $\pm$ 0.18 | $\pm$ 0.24 | $\pm$ 0.03 | $\pm$ 1.16 | $\pm$ 0.05 |
| | | | | | | DAL | | | | | | |
| #1 | 0.80 | 99.84 | 0.40 | 99.59 | 0.95 | 99.29 | 2.65 | 98.85 | 7.75 | 97.37 | 2.51 | 98.86 |
| #2 | 0.90 | 99.24 | 0.60 | 99.57 | 1.20 | 99.26 | 2.65 | 98.84 | 8.15 | 97.43 | 2.70 | 98.87 |
| #3 | 0.90 | 99.16 | 0.65 | 99.52 | 1.15 | 99.14 | 2.40 | 98.75 | 8.20 | 97.35 | 2.66 | 98.78 |
| #4 | 0.80 | 99.37 | 0.55 | 99.63 | 0.95 | 99.34 | 2.85 | 98.89 | 7.95 | 97.39 | 2.62 | 98.93 |
| #5 | 1.25 | 99.39 | 0.40 | 99.61 | 0.85 | 99.39 | 2.75 | 98.90 | 7.70 | 97.46 | 2.59 | 98.95 |
| mean | **0.93** | **99.40** | **0.52** | **99.58** | **1.02** | **99.28** | **2.65** | **98.84** | **7.95** | **97.40** | **2.61** | **98.87** |
| $\pm$ std | $\pm$ 0.17 | $\pm$ 0.24 | $\pm$ 0.10 | $\pm$ 0.04 | $\pm$ 0.13 | $\pm$ 0.08 | $\pm$ 0.15 | $\pm$ 0.05 | $\pm$ 0.20 | $\pm$ 0.04 | $\pm$ 0.06 | $\pm$ 0.06 |

summarize the experimental results in Tables 6-7. As we can see, our DAL can not only lead to improved performance in OOD detection, but our performance is also very stable across different choices of ID datasets and real OOD datasets.

### F.5 Effects of Hyper-parameters

We further test the impacts of other hyper-parameters on the performance in OOD detection, where we consider $\gamma_{\max}$, $\beta$, `num_search`, and `ps` on CIFAR benchmarks.

**Impacts of $\gamma$.** The exact values of $\gamma$ are dynamically determined by $\gamma_{\max}$, $\rho$, $\beta$, and the current model $\mathbf{f_w}$. To evaluation the effects of $\gamma$, we conduct experiments on CIFAR benchmarks with different $\gamma_{\max}$, $\rho$, and $\beta$, of which the results are summarized in Table 8-9. Overall, our method is pretty robust to different choices of hyper-parameters, in that the results for most of the hyper-parameter setups can lead to promising improvement over the original outlier exposure. Specifically, for $\gamma_{\max}$, most of its different choices can lead to effective OOD detection with the proper choices of $\rho$ and $\beta$, but its values should not be too small (e.g., $\gamma_{\max} = 0.1$). The reason is that if the value of $\gamma$ is too small, the distance between the worst-cases OOD features, i.e., $\mathbf{g_w}(\mathbf{x}) + \mathbf{p}$, and the original OOD features, i.e., $\mathbf{g_w}(\mathbf{x})$, can be very long, occupying the regions that should belong to ID data. It will make the model confused between ID and OOD cases and thus lead to unsatisfactory results. A similar conclusion can also be applied for $\beta$: when its value is too large (such as $\beta = 5$), values of $\mathbf{g_w}(\mathbf{x}) + \mathbf{p}$ can also be arbitrarily large, making the current model hardly learn to discern ID and OOD patterns.

**Impacts of `num_search` and `ps`.** We also provides the results on CIFAR benchmarks with different `num_search` and `ps`, and the results can be found in Tables 10-11. As we can see, even with some extreme values, such as `num_search` $= 500$ and `ps` $= 100$, the resultant models still enjoy the improvements over outlier exposure, indicating that our method is pretty robust to these hyper-parameters. The reason is that our proper selection of $\rho$ will constrain the resultant perturbation to be beneficial, avoiding the worst OOD distribution to not intrude the region that belongs to ID data.

Table 7: Comparison of DAL and outlier exposure on CIFAR-100 with 5 individual trails. ↓ (or ↑) indicates smaller (or larger) values are preferred, and a bold font indicates the best results in the corresponding column.

| Trails | SVHN FPR95↓ | SVHN AUROC↑ | LSUN FPR95↓ | LSUN AUROC↑ | iSUN FPR95↓ | iSUN AUROC↑ | Textures FPR95↓ | Textures AUROC↑ | Places365 FPR95↓ | Places365 AUROC↑ | Average FPR95↓ | Average AUROC↑ |
|---|---|---|---|---|---|---|---|---|---|---|---|---|
| | | | | | | OE | | | | | | |
| #1 | 44.45 | 91.76 | 15.75 | 97.26 | 45.95 | 88.80 | 47.35 | 89.80 | 54.10 | 87.90 | 41.52 | 91.10 |
| #2 | 42.75 | 91.93 | 15.85 | 97.22 | 46.85 | 88.91 | 46.75 | 89.78 | 53.05 | 88.04 | 41.05 | 91.18 |
| #3 | 43.75 | 91.88 | 15.95 | 97.34 | 52.25 | 87.62 | 47.15 | 89.49 | 54.10 | 88.03 | 42.64 | 90.87 |
| #4 | 41.30 | 92.23 | 16.15 | 97.27 | 46.90 | 88.76 | 47.00 | 89.73 | 54.40 | 87.91 | 41.15 | 91.18 |
| #5 | 42.55 | 91.92 | 16.20 | 97.22 | 44.70 | 89.66 | 47.35 | 89.47 | 54.35 | 87.82 | 41.03 | 91.22 |
| mean | 42.96 | 91.94 | 15.97 | 97.26 | 47.33 | 88.75 | 47.12 | 89.65 | 54.00 | 87.94 | 41.47 | 91.10 |
| ± std | ± 1.07 | ± 0.15 | ± 0.17 | ± 0.04 | ± 2.58 | ± 0.65 | ± 0.23 | ± 0.14 | ± 0.49 | ± 0.08 | ± 0.61 | ± 0.13 |
| | | | | | | DAL | | | | | | |
| #1 | 19.35 | 96.21 | 16.05 | 96.78 | 26.05 | 94.23 | 37.60 | 91.57 | 49.35 | 88.81 | 29.68 | 93.52 |
| #2 | 22.65 | 95.55 | 16.30 | 96.73 | 26.35 | 94.23 | 36.20 | 91.91 | 48.50 | 88.74 | 30.00 | 93.43 |
| #3 | 20.15 | 96.15 | 16.20 | 96.91 | 29.85 | 93.55 | 37.85 | 91.60 | 47.90 | 88.95 | 30.39 | 93.43 |
| #4 | 14.50 | 96.72 | 16.75 | 96.58 | 33.75 | 92.68 | 37.60 | 91.63 | 49.70 | 88.80 | 30.46 | 93.28 |
| #5 | 22.70 | 95.90 | 15.20 | 96.91 | 27.15 | 94.58 | 37.00 | 91.82 | 49.65 | 88.73 | 30.34 | 93.59 |
| mean | **19.87** | **96.11** | **16.10** | 96.78 | **28.63** | 93.85 | **37.25** | 91.70 | **49.01** | 88.80 | **30.17** | 93.45 |
| ± std | ± 2.99 | ± 0.39 | ± 0.51 | ± 0.12 | ± 2.89 | ± 0.68 | ± 0.60 | ± 0.13 | ± 0.71 | ± 0.08 | ± 0.29 | ± 0.10 |

Table 8: Detection Performance on CIFAR-10 dataset with different choices of $\beta$, $\rho$, and $\gamma_{\max}$, where we report the FPR95 / AUROC for each individual trail setup.

$\gamma_{\max}=50$

| $\beta$ \ $\rho$ | 1e-2 | 1e-1 | 1 | 10 | 100 |
|---|---|---|---|---|---|
| 1e-3 | 2.95 / 99.07 | 2.80 / **99.07** | 2.96 / 99.02 | 2.80 / 98.31 | 91.85 / 64.43 |
| 5e-3 | 2.95 / 99.01 | 3.05 / 99.00 | 2.97 / 99.04 | 2.69 / 98.16 | 92.79 / 55.39 |
| 1e-2 | 2.79 / 98.95 | 2.68 / 99.05 | 2.84 / 98.88 | 2.71 / 98.67 | 96.48 / 44.97 |
| 5e-2 | 3.08 / 98.98 | 3.03 / 98.98 | 2.85 / 98.98 | 2.88 / 98.79 | 95.58 / 45.79 |
| 1e-1 | 2.79 / 98.96 | 2.98 / 98.99 | 2.75 / 99.02 | 10.22 / 96.37 | 88.96 / 72.68 |
| 5e-1 | 2.82 / 99.00 | 2.95 / 99.01 | 2.81 / 99.05 | 4.34 / 96.58 | 95.51 / 46.60 |
| 1 | 2.94 / 98.93 | 2.88 / 99.02 | 3.19 / 99.00 | 52.76 / 94.36 | 95.05 / 53.89 |
| 5 | 2.98 / 98.91 | 2.77 / 98.96 | 3.00 / 99.06 | 94.44 / 62.90 | 95.57 / 52.51 |

$\gamma_{\max}=10$

| $\beta$ \ $\rho$ | 1e-2 | 1e-1 | 1 | 10 | 100 |
|---|---|---|---|---|---|
| 1e-3 | 3.06 / 99.05 | 2.82 / 99.06 | 2.84 / 98.97 | **2.41** / 97.95 | 94.65 / 46.60 |
| 5e-3 | 2.85 / 99.00 | 2.80 / **99.09** | 2.93 / 99.04 | 2.56 / 98.21 | 94.69 / 57.61 |
| 1e-2 | 2.91 / 98.98 | 2.98 / 99.04 | 2.56 / 99.02 | 2.58 / 98.28 | 95.48 / 50.70 |
| 5e-2 | 2.94 / 99.01 | 2.91 / 99.00 | 2.71 / 99.03 | 2.81 / 98.53 | 97.80 / 43.86 |
| 1e-1 | 3.02 / 99.04 | 2.77 / 99.04 | 2.87 / 99.02 | 14.11 / 95.47 | 96.00 / 55.75 |
| 5e-1 | 2.90 / 98.98 | 2.89 / 99.04 | 2.73 / 99.06 | 67.48 / 90.50 | 94.64 / 56.57 |
| 1 | 2.81 / 98.96 | 2.89 / 99.04 | 2.82 / 99.04 | 90.97 / 70.65 | 87.12 / 56.72 |
| 5 | 2.73 / 98.98 | 2.90 / 99.03 | 14.04 / 95.64 | 93.01 / 45.50 | 88.61 / 69.65 |

$\gamma_{\max}=5$

| $\beta$ \ $\rho$ | 1e-2 | 1e-1 | 1 | 10 | 100 |
|---|---|---|---|---|---|
| 1e-3 | 3.05 / 99.04 | 2.84 / 98.98 | 2.81 / 98.99 | 2.79 / 98.20 | 93.77 / 64.09 |
| 5e-3 | 2.82 / 99.02 | 2.85 / **99.20** | 2.97 / 99.00 | 2.76 / 98.37 | 93.50 / 63.44 |
| 1e-2 | 2.82 / 98.99 | 2.95 / 98.92 | 2.93 / 99.03 | 2.68 / 98.48 | 94.42 / 52.51 |
| 5e-2 | 2.99 / 98.98 | 2.81 / 98.99 | 2.87 / 98.91 | 5.21 / 95.95 | 87.32 / 71.25 |
| 1e-1 | 2.91 / 98.95 | 2.70 / 99.06 | 2.90 / 99.03 | 3.84 / 96.46 | 95.17 / 57.61 |
| 5e-1 | 3.03 / 99.01 | 3.06 / 99.00 | 2.75 / 99.02 | 88.76 / 69.56 | 94.68 / 41.66 |
| 1 | 2.67 / 99.00 | 2.86 / 99.03 | 2.94 / 99.00 | 97.40 / 45.45 | 95.03 / 48.58 |
| 5 | 2.87 / 98.93 | 2.82 / 98.97 | 63.22 / 87.13 | 98.97 / 58.50 | 95.49 / 50.51 |

$\gamma_{\max}=1$

| $\beta$ \ $\rho$ | 1e-2 | 1e-1 | 1 | 10 | 100 |
|---|---|---|---|---|---|
| 1e-3 | 2.74 / 98.92 | **2.62** / 98.95 | 2.64 / 98.93 | 2.70 / 98.93 | 92.36 / 57.40 |
| 5e-3 | 2.64 / 99.00 | 2.78 / 98.98 | 2.86 / 98.82 | 2.68 / 98.16 | 94.72 / 58.59 |
| 1e-2 | 2.69 / 99.02 | 2.69 / 99.00 | 2.69 / 98.97 | 2.79 / 98.19 | 93.39 / 57.55 |
| 5e-2 | 2.67 / 98.94 | 2.75 / 98.93 | 2.71 / 98.97 | 2.74 / 98.64 | 94.60 / 52.14 |
| 1e-1 | 2.91 / 98.99 | 2.71 / **99.07** | 2.77 / 98.96 | 22.93 / 94.15 | 92.55 / 67.68 |
| 5e-1 | 2.66 / 98.99 | 2.64 / 98.99 | 2.90 / 98.87 | 79.75 / 89.56 | 93.76 / 55.94 |
| 1 | 2.78 / 99.01 | 2.98 / 98.94 | 2.56 / 98.96 | 52.66 / 90.50 | 92.57 / 90.03 |
| 5 | 2.68 / 99.03 | 2.68 / 98.96 | 2.69 / 98.96 | 25.31 / 94.53 | 97.42 / 43.10 |

$\gamma_{\max}=0.5$

| $\beta$ \ $\rho$ | 1e-2 | 1e-1 | 1 | 10 | 100 |
|---|---|---|---|---|---|
| 1e-3 | 2.90 / 98.99 | 2.74 / 98.89 | 2.74 / 98.87 | 2.56 / 98.20 | 94.75 / 65.65 |
| 5e-3 | 2.64 / 98.94 | 2.79 / 98.89 | 2.53 / 98.81 | 2.54 / 98.32 | 97.22 / 41.88 |
| 1e-2 | 2.65 / 98.93 | 2.74 / 98.94 | 2.78 / 98.85 | 2.70 / 98.50 | 94.86 / 56.57 |
| 5e-2 | 2.68 / **98.98** | 2.73 / 98.95 | 2.89 / 98.80 | 3.50 / 96.78 | 92.13 / 65.63 |
| 1e-1 | 2.68 / 98.94 | 2.63 / 98.91 | 2.89 / 98.86 | 11.31 / 96.11 | 91.44 / 70.20 |
| 5e-1 | 3.07 / 98.89 | 2.74 / 98.89 | 2.50 / 98.68 | 18.10 / 95.47 | 93.15 / 59.78 |
| 1 | 2.82 / 98.85 | 2.61 / 98.89 | 2.76 / 98.81 | 12.58 / 95.80 | 94.61 / 61.49 |
| 5 | 2.61 / 98.84 | 2.73 / 98.92 | 2.74 / 98.73 | 83.18 / 90.90 | 94.14 / 60.82 |

$\gamma_{\max}=0.1$

| $\beta$ \ $\rho$ | 1e-2 | 1e-1 | 1 | 10 | 100 |
|---|---|---|---|---|---|
| 1e-3 | 2.75 / **98.59** | 53.32 / 94.12 | 82.19 / 92.03 | 2.52 / 98.41 | 95.24 / 51.41 |
| 5e-3 | 2.83 / 98.55 | **2.46** / 98.55 | 91.80 / 59.84 | 90.23 / 70.99 | 95.65 / 69.73 |
| 1e-2 | 2.47 / 98.48 | 2.56 / 98.55 | 87.64 / 88.24 | 97.56 / 71.35 | 88.87 / 77.29 |
| 5e-2 | 91.10 / 91.70 | **2.46** / 98.53 | 87.24 / 90.28 | 90.04 / 81.11 | 96.49 / 44.84 |
| 1e-1 | 87.87 / 91.66 | 3.00 / 95.54 | 86.08 / 71.34 | 89.70 / 72.59 | 73.47 / 79.13 |
| 5e-1 | 90.51 / 84.74 | 92.42 / 92.06 | 91.78 / 90.73 | 89.48 / 81.07 | 94.27 / 59.21 |
| 1 | 93.82 / 89.51 | 92.29 / 90.44 | 96.46 / 83.90 | 79.04 / 86.78 | 94.06 / 50.42 |
| 5 | 80.22 / 91.03 | 87.43 / 89.21 | 91.98 / 59.04 | 91.55 / 63.98 | 94.18 / 61.55 |

Table 9: Detection Performance on CIFAR-100 dataset with different choices of $\beta$, $\rho$, and $\gamma_{\max}$, where we report the FPR95 / AUROC for each individual trail setup.

| $\gamma_{\max}=50$ | | $\rho$ | | | | | $\gamma_{\max}=10$ | | $\rho$ | | | |
|---|---|---|---|---|---|---|---|---|---|---|---|---|
| | | 1e-2 | 1e-1 | 1 | 10 | 100 | | 1e-2 | 1e-1 | 1 | 10 | 100 |
| | 1e-3 | 31.72/93.37 | 30.84/93.51 | 31.50/93.27 | 31.08/92.47 | 87.70/86.84 | 1e-3 | 33.07/92.92 | 33.93/92.85 | 35.09/92.44 | 30.39/92.59 | 94.25/48.00 |
| | 5e-3 | 33.36/92.70 | 30.23/93.61 | 31.27/93.32 | 29.98/92.94 | 96.98/36.34 | 5e-3 | 34.03/92.81 | 33.64/92.88 | 30.43/93.66 | 33.75/92.00 | 96.58/48.51 |
| | 1e-2 | 31.73/93.32 | 35.42/92.42 | 30.86/93.61 | 31.22/92.34 | 87.17/86.80 | 1e-2 | 35.80/92.46 | 33.40/93.16 | 34.52/92.90 | 32.02/91.76 | 88.11/86.65 |
| $\beta$ | 5e-2 | 33.69/92.66 | 32.50/93.18 | 31.29/93.40 | 30.92/92.68 | 87.23/86.57 | 5e-2 | 33.38/93.19 | 32.95/93.29 | 29.74/**93.63** | 29.13/92.71 | 95.18/50.19 |
| | 1e-1 | 32.21/93.18 | 32.42/93.31 | 33.77/92.98 | 29.48/93.04 | 87.14/87.27 | 1e-1 | 31.77/93.33 | 31.94/93.31 | 35.34/92.78 | **27.80**/93.02 | 95.48/56.81 |
| | 5e-1 | 33.44/93.16 | 36.19/92.71 | 33.55/92.85 | 34.83/91.70 | 86.91/87.22 | 5e-1 | 34.12/92.71 | 35.26/92.86 | 31.71/93.23 | 79.79/89.56 | 89.26/86.60 |
| | 1 | 33.01/93.16 | 33.80/92.64 | **28.99/93.87** | 36.22/91.41 | 92.34/52.72 | 1 | 32.90/93.05 | 34.13/92.86 | 31.64/93.05 | 47.37/91.31 | 96.10/46.88 |
| | 5 | 34.62/92.55 | 32.90/93.15 | 36.00/92.14 | 95.59/89.09 | 93.69/96.87 | 5 | 32.65/93.13 | 34.41/92.86 | 33.02/91.80 | 57.19/90.72 | 89.12/87.75 |
| $\gamma_{\max}=5$ | | $\rho$ | | | | | $\gamma_{\max}=1$ | | $\rho$ | | | |
| | | 1e-2 | 1e-1 | 1 | 10 | 100 | | 1e-2 | 1e-1 | 1 | 10 | 100 |
| | 1e-3 | 33.58/92.84 | 34.61/92.35 | 32.90/92.91 | 30.55/92.66 | 95.80/54.59 | 1e-3 | 37.24/91.87 | 30.42/93.44 | 33.45/92.88 | 30.17/92.77 | 95.60/57.62 |
| | 5e-3 | 33.80/92.87 | 34.98/92.61 | 31.87/93.39 | 27.36/93.18 | 89.37/51.52 | 5e-3 | 32.75/92.88 | 32.24/93.05 | 32.13/92.79 | 31.37/92.68 | 95.96/48.31 |
| | 1e-2 | 36.96/92.48 | 34.87/93.01 | 30.87/93.01 | 31.75/92.82 | 85.67/88.23 | 1e-2 | 36.66/91.81 | 30.40/93.45 | 29.47/93.54 | 31.14/92.37 | 88.33/86.39 |
| $\beta$ | 5e-2 | 36.00/92.49 | 32.10/93.24 | 31.31/**93.55** | 30.70/92.95 | 85.80/86.38 | 5e-2 | 31.00/93.39 | 30.88/**93.46** | 30.33/92.97 | 31.43/92.50 | 87.83/88.46 |
| | 1e-1 | 33.64/92.75 | 31.82/93.09 | 32.55/92.97 | 32.42/92.22 | 91.16/84.87 | 1e-1 | 31.18/93.42 | 33.03/93.08 | 31.71/93.17 | 52.46/90.95 | 95.97/53.86 |
| | 5e-1 | 34.03/92.69 | 33.03/93.09 | 32.99/93.19 | 81.64/89.24 | 89.67/87.08 | 5e-1 | 35.14/92.79 | 29.55/93.34 | **29.19**/93.34 | 82.19/89.22 | 89.30/85.40 |
| | 1 | 34.85/92.61 | 32.73/93.31 | 29.53/93.74 | 63.16/90.50 | 87.71/87.06 | 1 | 36.93/92.33 | 34.03/93.00 | 35.45/91.74 | 77.26/90.46 | 97.33/48.51 |
| | 5 | 34.59/92.69 | 33.14/93.16 | 72.10/90.79 | 40.65/91.30 | 89.45/86.70 | 5 | 31.00/93.37 | 30.48/93.37 | 31.17/93.21 | 83.14/89.83 | 97.43/50.74 |
| $\gamma_{\max}=0.5$ | | $\rho$ | | | | | $\gamma_{\max}=0.1$ | | $\rho$ | | | |
| | | 1e-2 | 1e-1 | 1 | 10 | 100 | | 1e-2 | 1e-1 | 1 | 10 | 100 |
| | 1e-3 | 33.10/92.59 | 32.25/92.84 | 30.40/92.97 | 31.70/92.39 | 87.47/86.53 | 1e-3 | 93.84/50.41 | 99.39/51.61 | 76.69/89.44 | 75.19/89.36 | 97.09/44.87 |
| | 5e-3 | 34.15/92.33 | 31.80/92.33 | 31.97/92.49 | 31.75/92.48 | 90.14/86.43 | 5e-3 | 81.55/87.48 | 80.81/88.63 | 75.63/90.32 | 96.69/43.59 | 96.15/50.92 |
| | 1e-2 | 34.35/92.28 | 33.90/92.29 | 29.72/**93.29** | 29.66/92.75 | 88.49/86.76 | 1e-2 | 95.32/51.45 | 91.92/55.05 | 75.40/89.75 | 96.55/45.81 | 95.24/48.11 |
| $\beta$ | 5e-2 | 32.61/93.04 | 33.19/92.41 | 33.73/92.23 | 30.28/92.99 | 84.51/88.00 | 5e-2 | **42.65**/90.84 | 93.80/45.56 | 99.58/50.06 | 97.08/45.02 | 83.93/87.53 |
| | 1e-1 | 33.23/92.48 | 35.78/91.90 | 33.89/91.97 | 75.36/90.70 | 96.66/57.33 | 1e-1 | 95.32/51.45 | 91.92/55.05 | 75.40/89.75 | 96.55/45.81 | 95.24/48.11 |
| | 5e-1 | 32.48/92.33 | 30.56/93.19 | 32.74/92.37 | 84.06/88.13 | 91.47/46.58 | 5e-1 | 90.51/84.74 | 92.42/**92.06** | 91.78/90.73 | 89.48/81.07 | 94.27/59.21 |
| | 1 | 33.90/92.25 | 32.89/92.52 | 31.26/92.89 | 62.99/90.64 | 88.53/87.17 | 1 | 81.14/90.18 | 82.34/89.98 | 80.38/90.42 | 88.00/53.88 | 95.43/46.75 |
| | 5 | 32.62/92.33 | 32.60/92.48 | 30.04/92.94 | 71.77/90.34 | 96.58/49.03 | 5 | 96.57/45.61 | 98.16/47.34 | 100.0/49.86 | 94.89/51.80 | 92.12/87.10 |

Table 10: The hyper-parameter effects of `num_search` on the CIFAR benchmarks.

| | 0 | 1 | 2 | 5 | 10 | 20 | 50 | 100 | 200 |
|---|---|---|---|---|---|---|---|---|---|
| **CIFAR-10** | | | | | | | | | |
| FPR95 | 3.33 | 2.90 | **2.41** | 2.61 | 2.62 | 2.46 | 2.74 | 2.86 | 3.00 |
| AUROC | 98.59 | **99.10** | 98.96 | 98.91 | 98.92 | 98.56 | 98.95 | 99.07 | 98.80 |
| **CIFAR-100** | | | | | | | | | |
| FPR95 | 36.47 | 34.12 | 33.55 | 33.30 | **30.38** | 31.27 | 32.01 | 33.07 | 31.73 |
| AUROC | 91.75 | 92.60 | 92.98 | 93.14 | **93.62** | 93.36 | 93.18 | 92.91 | 93.22 |

Table 11: The hyper-parameter effects of `ps` on the CIFAR benchmarks.

| | $1e^{-2}$ | $5e^{-2}$ | $1e^{-1}$ | $5e^{-1}$ | 1 | 5 | 10 | 50 | 100 |
|---|---|---|---|---|---|---|---|---|---|
| **CIFAR-10** | | | | | | | | | |
| FPR95 | 2.97 | 2.76 | 2.80 | **2.49** | 2.57 | 2.92 | 3.01 | 3.04 | 2.92 |
| AUROC | 99.00 | **99.02** | 98.95 | 98.94 | 98.82 | 98.90 | 98.97 | 98.81 | 98.30 |
| **CIFAR-100** | | | | | | | | | |
| FPR95 | 35.74 | 35.75 | 32.64 | **29.00** | 31.03 | 33.63 | 32.93 | 37.61 | 95.07 |
| AUROC | 92.82 | 92.45 | 93.14 | **93.95** | 93.18 | 92.74 | 92.74 | 93.09 | 91.17 |

### F.6 Aligning Training Epochs

In our experiments, we follow the suggested hyper-parameters for the baselines, running OE with 10 epochs on the CIFAR benchmarks. However, our DAL, due to distribution augmentation, is run for 50 epochs to fully fit the augmented distribution. To demonstrate that our improvement is not dominated by longer training time, we also list the results of OE with 50 epochs training, summarizing the results on the CIFAR benchmarks in Table 12. As we can see, although OE can produce better results with 50 epochs of training, our DAL can still demonstrate its superiority in OOD detection. For example, on CIFAR-100, our DAL improves OE by 7.67 w.r.t. FPR95 and 1.92 w.r.t. AUROC.

Table 12: Comparison between OE and DAL with 50 epochs training.

|     | CIFAR-10 | | CIFAR-100 | |
| --- | --- | --- | --- | --- |
|     | FPR95 ↓ | AUROC ↑ | FPR95 ↓ | AUROC ↑ |
| OE  | 3.07 | 98.97 | 37.35 | 92.00 |
| DAL | **2.68** | **99.01** | **29.68** | **93.92** |

### F.7 Other Norms

We can also use the $l_2$ norm and the associated Wasserstein-2 distance. Therefore, we conduct the related experiments on the CIFAR benchmarks in comparing between $l_1$ and $l_2$ norms, and the results are summarized in Table 13. As we can see, we do not observe an obvious difference between using $\ell_1$ and $\ell_2$ norms, so it is proper to use the $\ell_1$ norm and the Wasserstein-1 distance in our DAL by default.

Table 13: Using $\ell_1$ and $\ell_2$ norms.

|     | $\ell_1$ norm | | $\ell_2$ norm | |
| --- | --- | --- | --- | --- |
|     | FPR95 ↓ | AUROC ↑ | FPR95 ↓ | AUROC ↑ |
| CIFAR-10  | **2.68** | **99.01** | 2.81 | 98.98 |
| CIFAR-100 | **29.68** | 93.92 | 30.20 | **93.95** |

### F.8 Linear Probing

In many applications, the costs of re-training and re-deployment can be prohibitively high, where we should assume a fixed feature extractor **e** and allow only the classifier **h** (i.e., the fully connected layer) to be tuned. DAL is also adaptable for such a restricted setting, with improved detection performance over the OE counterpart. Table 14 summarizes the results on CIFAR-100, comparing OE and DAL under the settings of full training (fine tuning) and training with only the classifier (linear probe). As we can see, for the linear probe setup, DAL can still improve the OE counterpart, while the performance gain is largely limited compared to that of the full training.

Table 14: Comparison between fully fine-tuning and linear probing.

| FPR95 ↓ | linear probe | fine tune |
| --- | --- | --- |
| OE  | 50.09 | 43.14 |
| DAL | **43.37** | **29.68** |

### F.9 False Negative Rate

We further consider the metric of false negative rate (FNR95) for ID data when the true positive rate of ID data is at 95%. We summarize the results on the CIFAR benchmarks in Table 15, where we consider the common OOD detection setups as in Table 1 and the challenging CIFAR-10 vs. CIFAR-100 setup as in Table 2. As we can see, the FNR decreases for all three considered cases, further demonstrating the effectiveness of our method.

Table 15: Experiments measured by FNR95.

| FNR95 ↓ | CIFAR-10 | CIFAR-100 | CIFAR-10 vs. CIFAR-100 |
| --- | --- | --- | --- |
| MSP | 33.02 | 64.83 | 43.01 |
| OE  | 5.04 | 41.31 | 26.38 |
| DAL | **3.89** | **26.87** | **22.81** |

Table 16: Comparison between our method and advanced methods on ImageNet. ↓ (or ↑) indicates smaller (or larger) values are preferred, and a bold font indicates the best results in the column.

| Method | Textures | | Places365 | | iNaturalist | | SUN | | Average | |
|---|---|---|---|---|---|---|---|---|---|---|
| | FPR95↓ | AUROC↑ | FPR95↓ | AUROC↑ | FPR95↓ | AUROC↑ | FPR95↓ | AUROC↑ | FPR95↓ | AUROC↑ |
| Using ID data only | | | | | | | | | | |
| MSP | 66.58 | 80.03 | 74.15 | 78.97 | 72.72 | 77.19 | 78.70 | 75.15 | 73.04 | 77.84 |
| Free Energy | 52.84 | 86.36 | 70.64 | 81.67 | 73.98 | 75.97 | 76.92 | 78.08 | 68.60 | 80.52 |
| ASH | 15.93 | 96.00 | 63.08 | 82.43 | 52.05 | 83.67 | 71.68 | 77.71 | 50.68 | 85.35 |
| Mahalanobis | 40.52 | 91.41 | 97.10 | 53.11 | 96.15 | 53.62 | 96.95 | 52.74 | 82.68 | 62.72 |
| KNN | 26.54 | 93.49 | 78.64 | 76.82 | 75.78 | 69.51 | 74.30 | 78.85 | 63.82 | 79.66 |
| VOS | 94.83 | 57.69 | 98.72 | 38.50 | 87.75 | 65.65 | 70.20 | 83.62 | 87.87 | 61.36 |
| Using ID data and auxiliary OOD data | | | | | | | | | | |
| OE | 57.34 | 82.97 | 7.92 | 98.04 | 73.87 | 76.94 | 52.60 | 77.31 | 52.60 | 83.81 |
| Energy-OE | 42.46 | 88.27 | 1.88 | 99.49 | 73.81 | **78.34** | 69.45 | 79.54 | 46.90 | 86.41 |
| ATOM | 60.20 | 90.60 | 7.07 | 98.25 | 74.30 | 77.00 | 55.87 | 75.80 | 49.36 | 85.41 |
| DOE | 35.11 | 92.15 | 0.72 | 99.79 | 72.55 | 78.00 | 59.06 | 85.67 | 41.86 | 88.90 |
| POEM | 40.80 | 89.78 | 0.26 | 99.70 | 73.23 | 68.83 | 65.45 | 82.08 | 44.93 | 85.10 |
| DAL | 55.49 | 85.29 | 5.83 | 99.09 | 74.23 | 76.70 | 50.76 | 79.21 | 46.57 | 85.08 |
| DAL-ASH | **14.10** | **97.00** | **0.23** | **99.85** | **67.38** | 78.20 | **45.14** | **85.90** | **31.71** | **90.24** |
| DAL-Energy | 33.83 | 90.44 | 0.47 | 99.82 | 74.37 | 67.68 | 49.12 | 80.28 | 39.45 | 84.55 |

## F.10 ImageNet Evaluations

We also conduct experiments on the ImageNet benchmarks, demonstrating the effectiveness of our DAL when facing this very challenging OOD detection task.

**OOD Datasets.** We adopt a subset of ImageNet-21K-P dataset (Ridnik et al., 2021) as the auxiliary OOD data, which is cleansed to avoid those classes that coincide with ID cases. Furthermore, iNaturalist (Horn et al., 2018), SUN (Xu et al., 2015), Places365, and Textures are adopted as the real OOD datasets, where we eliminate those data whose labels coincide with ID cases.

**Hyper-parameter Selection.** The hyper-parameters are also tuned on the validation data. We adopt the random search that follows the following three steps. Step 1: we randomly select a hyper-parameter (e.g., $\beta$) and fix the values of all other hyper-parameters to be their current optimal values. Step 2: we choose the best $\beta$ from the candidate set. Step 3: do Steps 1-2 again. We repeat Steps 1 and 2 for 50 times in our experiments. For the backbone model, we use ResNet-50 with the pre-trained parameters published by the PyTorch official repository.

**Hyper-parameters Setups.** Our DAL is run for 5 epochs. The batch size is 64 for both the ID and the OOD cases. We have the initial learning rate $1e^{-4}$, $\gamma_{max} = 10$, $\beta = 0.5$, $\rho = 0.1$, and ps = 0.1. Furthermore, we employ cosine decay (Loshchilov and Hutter, 2017) for the learning rate.

**ImageNet evaluations.** Due to the large semantic space and complex image patterns, OOD detection on the ImageNet dataset is a challenging task (Huang and Li, 2021). However, similar to the CIFAR benchmarks, DAL can also reveal the best detection performance over all the considered baseline methods. Moreover, it is well-known that MSP scoring can easily fail on the ImageNet benchmark (Hendrycks et al., 2022), so we also report the results after DAL training using ASH (DAL-ASH) and Free Energy (DAL-Energy), which can further improve the detection performance.