# OpenReview forum: "Learning to Augment Distributions for Out-of-distribution Detection"
_NeurIPS.cc/2023/Conference — NeurIPS 2023 poster_

### Official Review · Reviewer_3p7h · 2023-06-27

**Soundness:** 4 excellent
**Presentation:** 4 excellent
**Contribution:** 4 excellent
**Rating:** 7
**Confidence:** 3

**Summary:**

This paper noticed that the discrepancy between the auxiliary and (unseen) real OOD data hindered the effectiveness of outlier exposure, which is a potential method for OOD detection. To tackle this issue, the authors proposed a learning framework that augments the auxiliary OOD dataset using a Wasserstein ball. By training the model on the worst-case OOD data in the ball, the performance of the model toward open-world OOD detection will be improved.

**Strengths:**

- The paper is well-motivated: the authors formalized the challenges in existing outlier exposure methods for OOD detection and provide a solution to the challenges.
- The proposed DAL framework is proposed with firm theoretical support with a convergence guarantee.
- Experiments show that using ID and OOD auxiliary samples and using the proposed DAL can effectively improve the performance of OOD detection, even when ID and real OOD samples are very close. The results confirm the design of DAL.

**Weaknesses:**

- Experiments are conducted on relatively simple datasets and models. It is worth to be shown that the proposed method still has improvements on more complicated samples.

**Questions:**

- Given the Wasserstein ball is applied to the feature distribution, is it possible to train the classifier only and leave the encoder as pre-trained?
- Will the false negative rate of ID samples be increased under DAL, i.e., ID samples being misclassified as OOD samples, especially in hard OOD detections?
- Why choose 10 searches in experiments? Will increased searches improve performance? How long does it take for the DAL training?

**Limitations:**

The authors do not address the limitations of the method. Possible improvements can be: can the model be easily applied to (1) regression tasks? (2) models that are hard to be fully fine-tuned, i.e., transformers.

---

> ### Author Rebuttal · Authors · 2023-08-08
>
> e sincerely thank you for your constructive comments and generous supports! Please find our responses below.
>
> > Q1. It is worth to be shown that the proposed method still has improvements on more complicated samples.
>
> A1. We have tested our DAL for some challenging setups in the literature [1,2], and some of the results can be found in the Appendix. For example, we evaluate on hard OOD detection in Section 5.2 and on ImageNet benchmark in Appendix F. In the future, we will contribute to the community to collecting new datasets for more complicated OOD detection evaluations.
>
> > Q2. Given the Wasserstein ball is applied to the feature distribution, is it possible to train the classifier only and leave the encoder as pre-trained?
>
> A2. Yes, we can fine-tune only the fully connected layer. However, the performance gain for OOD detection is more or less limited. Since traditionally trained models (e.g., cross-entropy loss on ID data) possess misleading representations that can degrade OOD detection, training only with the fully connected layer is insufficient. We conduct experiments on CIFAR-100 to support the above claims, summarized in the following table (fully training as "fine tuning" and training only with the FC layer as "linear probe"). As we can see, for the linear probe setup, DAL can still improve the OE counterpart, while the performance gain is largely limited compared to that of the fully training. We will add the related discussion in the Appendix.
>
> | FPR-95 | Linear Probe | Fine Tune |
> |:------:|:------------:|:---------:|
> |   OE   |     50.09    |   **43.14**   |
> |   DAL  |     43.37    |   **29.68**   |
>
> > Q3. Will the false negative rate of ID samples be increased under DAL, i.e., ID samples being misclassified as OOD samples, especially in hard OOD detections?
>
> A3. Many thanks for your constructive comments. We list the results of common OOD detection on the CIFAR benchmarks and the challenging CIFAR-10 vs. CIFAR-100 setup. The results are summarized in the following table. As we can see, the FNR decreases for all the three considered cases, further demonstrating the effectiveness of our method.
>
> | FNR-95 | CIFAR-10 common | CIFAR-100 common | CIFAR-10 vs. CIFAR-100 |
> |:------:|:---------------:|:----------------:|:----------------------:|
> |   MSP  |      33.02      |       64.83      |          43.01         |
> |   OE   |       5.04      |       41.31      |        26.38      |
> |   DAL  |    **3.89**    |     **26.87**   |          **22.81**         |
>
>
>
> > Q4. Why choose 10 searches in experiments? Will increased searches improve performance? How long does it take for the DAL training?
>
> A4. num_search} is a hyper-parameter that validation sets can tune. In Appendix F.5, we validate DAL across various num_search, finding that too larger values of  num_search can degrade the performance and num_search is a proper choice.
>
> Moreover, since data are augmented in the embedding space, each search step only takes one layer of backward propagation. Even with large values (e.g., num_search=50), its computation cost is still negligible compared to model updates.
>
>
> > Q5. The authors do not address the limitations of the method. Possible improvements can be: can the model be easily applied to (1) regression tasks? (2) models that are hard to be fully fine-tuned, i.e., transformers.
>
> A5. We discussed the limitation of our method in Appendix E.2, where the trade-off between the estimation error and the approximation error may limit the further improvement of our method. Moreover, other data augmentation approaches, which lead to complex augmentation or consider semantic diversity, are also open challenges.
>
> Applying our method to regression tasks and transformer fine-tuning is also attractive, while seldom work has studied these challenging setups. Some seminal works may include [3,4]. We will add the related discussion in the Appendix. Moreover, how to generalize our method for other problem setups in machine learning is also interesting.
>
> -----------------------
>
> [1] Yiyou Sun, Yifei Ming, Xiaojin Zhu, and Yixuan Li. "Out-of-distribution detection with deep nearest neighbors." ICML, 2022.
>
> [2] Rui Huang and Yixuan Li. "Mos: Towards scaling out-of-distribution detection for large semantic space." CVPR, 2021.
>
> [3] Geoff Pleiss, Amauri Souza, Joseph Kim, Boyi Li, and Kilian Q. Weinberger. "Neural network out-of-distribution detection for regression tasks." 2019.
>
> [4] Yunhao Ge, Jie Ren, Andrew Gallagher, Yuxiao Wang, Ming-Hsuan Yang, Hartwig Adam, Laurent Itti, Balaji Lakshminarayanan, and Jiaping Zhao. "Improving Zero-shot Generalization and Robustness of Multi-modal Models." CVPR, 2023.

---

> > ### Comment · Reviewer_3p7h · 2023-08-15
> >
> > The authors have successfully addressed my concerns. Thus, I will keep my score unchanged.

---

### Official Review · Reviewer_UEQu · 2023-07-04

**Soundness:** 3 good
**Presentation:** 3 good
**Contribution:** 3 good
**Rating:** 7
**Confidence:** 4

**Summary:**

This paper proposes a theory and an algorithm to address the OOD detection problem better. The whole paper is well-structured, and the algorithm is designed via the developed theory. The basic idea is to simulation the OOD distributions as representative as possible, thus we can expect that the detector can work well in the most scenarios. Experiments verify the effectiveness of the proposed method. All theoretical results are proved in the appendix.

**Strengths:**

1. This paper's structure is very clear, and readers enjoy reading this paper. It starts from a classical but powerful solution of OOD detection problem. Then, a theory follows to demonstrate the effectiveness of this kind of solution. Finally, am algorithm is designed via the proposed theory.

2. The proposed algorithm is verified in many benchmark datasets, including hard OOD situation, which is a strong evidence to support the claims made in this paper.

3. The studied problem is very important, and the focused research direction is practical. Although there are many non-outlier methods used to detect OOD data, we can always obtain some outliers to help our training, which is practical.


**Weaknesses:**

1. There might be some gaps between the algorithm and theory. One might be addressed during the response phase: In Theorem 2, there are two assumptions regarding the loss function. How are the both assumptions satisfied in your algorithm? Some discussions should be added here.

2. In lines 218-219, the reason why we need to perturb data in the embedding space should be clarified. You indeed follow previous work, but more explanations are needed here.

3. In lines 228-235, more justifications should be added here. I did not see any support (from literature or experiments) for this kind of design.

4. Experiments regarding CIFAR-10 can be moved to appendix, as all methods have good performance. It is not meaningful to see that one method is better than the other by 0.2%. In this way, more experiments in appendix can be moved back to the main content.

5. It is very interesting to see the real validation set here, more details can be included in the main content.


**Questions:**

See the weakness.

---

> ### Author Rebuttal · Authors · 2023-08-08
>
> We sincerely thank you for your constructive comments and generous supports! Please find our responses below.
>
>
> > Q1. In Theorem 2, there are two assumptions regarding the loss function. How are the both assumptions satisfied in your algorithm? Some discussions should be added here.
>
> A1. One can prove that the cross-entropy and the exponential losses are bounded and lipschitz w.r.t. $\mathbf{w}$ for deep models with softmax outputs [1], if
>
> - (1) activation functions are 1-Lipschitz;
> - (2) inputs are from bounded feature space $\mathcal{X}$;
> - (3) parameter space $\mathcal{W}$ is bounded (e.g., using regularization to ensure).
>
> More specifically, when the F-norm bounds parameters, softmax output $\mathbf{f}$ is continuous and never attains infinity. If we further assume that inputs are from a bounded feature space, then $\mathbf{f}$ is a continuous function over the bounded space, implying that $\mathbf{f}$ has upper and lower bounds. **Thus the cross-entropy and the exponential loss can be bounded in practice and our assumptions are practical.**
>
>
> > Q2. In lines 218-219, the reason why we need to perturb data in the embedding space should be clarified. You indeed follow previous work, but more explanations are needed here.
>
> A2.  **Computational efficiency is our primary consideration**. Taking ten searching steps for data augmentation, computing in the input space (ten times of full backward propagations per step) takes about ten times as long as that in the embedding space. Moreover, directly optimizing input features can easily get stacked at local solutions [3] and may require complex generative models [4], leading to unnecessary troubles.
>
> > Q3. In lines 228-235, more justifications should be added here. I did not see any support (from literature or experiments) for this kind of design.
>
> A3. We follow previous works that using min-max learning schemes in deep learning, e.g., [4,5]. Such an end-to-end realization is easy to follow and efficient in practice, which motivates our stochastic realization of DAL.
>
> > Q4. Experiments regarding CIFAR-10 can be moved to appendix, as all methods have good performance. It is not meaningful to see that one method is better than the other by 0.2%. In this way, more experiments in appendix can be moved back to the main content.
>
> A4. Sincere thanks for your suggestion. We will re-organize the experimental parts in our revision, moving the CIFAR-10 results to the Appendix.
>
> > Q5. It is very interesting to see the real validation set here, more details can be included in the main content.
>
> A5. As described in Section 5, **our hyper-parameters are tuned based on the validation data, separated from the training ID and auxiliary OOD data**. For example, for CIFAR-10, the validation data consist of 10K images from training set of CIFAR-10 and 10K images from Tiny-ImageNet-200 (i.e., auxiliary OOD data).
>
> Selecting a part of training data for validation is a common practice in machine learning, and such a tuning strategy is also used in many previous works [6,7]. We will discuss more about the validation set in our revision.
>
> ------------------------
>
> [1] Noah Golowich, Alexander Rakhlin, and Ohad Shamir. "Size-independent sample complexity of neural networks." COLT, 2018.
>
> [2] Qizhou Wang, Feng Liu, Bo Han, Tongliang Liu, Chen Gong, Gang Niu, Mingyuan Zhou, and Masashi Sugiyama. "Probabilistic margins for instance reweighting in adversarial training." NeurIPS, 2021.
>
> [3] Yuheng Zhang, Ruoxi Jia, Hengzhi Pei, Wenxiao Wang, Bo Li, and Dawn Song. "The secret revealer: Generative model-inversion attacks against deep neural networks." CVPR, 2020.
>
> [4] Akshay Mehra, Bhavya Kailkhura, Pin-Yu Chen, and Jihun Hamm. "On Certifying and Improving Generalization to Unseen Domains." 2022.
>
> [5] Tuan Anh Bui, Trung Le, Quan Tran, He Zhao, and Dinh Phung. "A unified wasserstein distributional robustness framework for adversarial training." ICLR, 2022.
>
> [6] Yiyou Sun, Yifei Ming, Xiaojin Zhu, and Yixuan Li. "Out-of-distribution detection with deep nearest neighbors." ICML, 2022.
>
> [7] Dan Hendrycks, Mantas Mazeika, and Thomas Dietterich. "Deep anomaly detection with outlier exposure." ICLR, 2019.

---

### Official Review · Reviewer_xghg · 2023-07-05

**Soundness:** 3 good
**Presentation:** 3 good
**Contribution:** 3 good
**Rating:** 6
**Confidence:** 3

**Summary:**

This paper aims to address the outlier exposure in OOD detection from theoretical perspective. By developing a novel generalization bound based on the distributional augment principle, the paper proposes a novel outlier exposure method. Experiments indicate the effectiveness of the proposed method.

**Strengths:**

1. A novel generalization bound to understand outlier exposure. It seems this theory is the first theory in outlier exposure field.
2. The proposed algorithm is theoretical guided and guarantee. From my personal view, this is important  to reliable ML.
3. The features trained based on proposed DAL can be used with other post-hoc OOD detection method. This method can be combined with previous works well and naturally.
4. Extensive experiments have shown the effectiveness of the method. It seems the performance has achieved SOTA performance compared with latest works.

**Weaknesses:**

1. I am confused that it seems the radius rho becomes larger, the estimation error in Theorem 3 will decrease. So, from the theoretical view, why not selected rho larger enough?
2. How to tune the parameters rho, gamma….? Validate set is allowed in outlier exposure?
3. It seems your algorithm and theory have a small gap. Could you clarity why you do that? It is better to give more clear explanations.

**Questions:**

Please answer the questions in Weakness.

---

> ### Author Rebuttal · Authors · 2023-08-08
>
> We sincerely thank you for your constructive comments and generous supports! Please find our responses below.
>
> > Q1. I am confused that it seems the radius rho becomes larger, the estimation error in Theorem 3 will decrease. So, from the theoretical view, why not selected rho larger enough?
>
> A1. As demonstrated in Theorem 3 and Figure 3, **there is a trade-off between the approximation and estimation errors**. Although a larger $\rho$ leads to a closer bound (i.e., a smaller estimation error), the approximate risk will become larger. Therefore, we should choose a proper $\rho$ to balance the approximation and estimation errors.
>
>
> > Q2. How to tune the parameters rho, gamma? Validate set is allowed in outlier exposure?
>
> A2. As described in Section 5, **our hyper-parameters are tuned based on validation data, separated from the training ID and auxiliary OOD data**. In fact, outlier exposure [1] valso adjusts the hyper-parameters with such validation data, stating that "the $\lambda$ coefficients were determined early in experimentation with validation $\mathcal{D}_\text{out}^\text{val}$ distributions described in Appendix A." Therefore, using the validation data to tune the hyper-parameters in OOD detection is proper.
>
>
> > Q3. It seems your algorithm and theory have a small gap. Could you clarity why you do that? It is better to give more clear explanations.
>
> A3. Perturbing features in the embedding space in Algorithm 1 leads the gap between our realization and theories. However, when applying our theories to the embedding space, the gap can be addressed.
>
> We would like to further explain that **computational efficiency is our primary consideration in using embedding perturbation**. With ten searching steps for data augmentation, computing in the input space (ten times of full backward propagations per step) takes about ten times as long as that in the embedding space. Moreover, directly optimizing input features can easily get stacked at local solutions [2] and may require complex generative models [3], leading to unnecessary troubles.
>
> --------------------------------
>
> [1] Dan Hendrycks, Mantas Mazeika, and Thomas Dietterich. "Deep anomaly detection with outlier exposure." ICLR, 2019.
>
> [2] Qizhou Wang, Feng Liu, Bo Han, Tongliang Liu, Chen Gong, Gang Niu, Mingyuan Zhou, and Masashi Sugiyama. "Probabilistic margins for instance reweighting in adversarial training." NeurIPS, 2021.
>
> [3] Yuheng Zhang, Ruoxi Jia, Hengzhi Pei, Wenxiao Wang, Bo Li, and Dawn Song. "The secret revealer: Generative model-inversion attacks against deep neural networks." CVPR, 2020.

---

### Official Review · Reviewer_REz1 · 2023-07-08

**Soundness:** 3 good
**Presentation:** 3 good
**Contribution:** 3 good
**Rating:** 7
**Confidence:** 2

**Summary:**

The paper aims to tackle the problem of out-of-distribution (OOD) detection in supervised learning through a learning theoretic approach. The authors propose a method termed DAL, which intends to reduce the discrepancy between the detector's training distribution (auxiliary OOD distribution) and true OOD distribution by expanding the set of auxiliary distributions over which the detector is trained. The paper outlines a rigorous approach to open-world OOD detection. DAL is evaluated experimentally and shown to outperform competing techniques.

Post-rebuttal update: increasing score to 7

**Strengths:**

Strengths:
1. The authors recognize the shortcomings in prior work on OOD detection and motivate the problem towards their proposed solution
2. While I am not an expert in the area, using Wasserstein type metrics and controlling the distribution discrepency between true and auxiliary OOD distributions appeals to mean
3. Through Theorems 2 and 3, the paper shows that DAL has a theoretical grounding and gives an algorithm to achieve the DAL objective
4. While I have not checked the proofs in sufficient detail, I believe that the analysis is rigorous
5. The experiments show the improvements achieved by DAL over several recent competing approaches

**Weaknesses:**

Weaknesses:
1. Theorems 2 and 3 can benefit from a detailed discussion in the main body
2. This could be a question as well - are common corruptions of CIFAR10 or Imagenet not used as standard datasets for OOD detection?
Also, please comment on my questions below

**Questions:**

Questions:
1. Line 88: Does this mean that the labels in ID and OOD are mutually exclusive? Could you please explain if this is a significant assumption
2. Line 124: Please provide references for the statement "the auxiliary and the real OOD data differ largely in practice". I would like to understand how significant is the typical difference.
3. As a follow-up, how large can the upper bound in (4) grow? Are there any standard assumptions made for theoretical analysis?
4. What does the assumption in Theorem 2 regarding l <= M imply- suppose l is cross-entropy or exponential loss, does the bound mean that f_w needs to be a reasonably good solution? Otherwise, the loss function could be unbounded, if I am not mistaken. How mild are these assumptions?

**Limitations:**

It would be useful if the authors discussed a few most relevant open challenges, in light of their findings

---

> ### Author Rebuttal · Authors · 2023-08-08
>
> We sincerely thank you for your constructive comments and generous supports! Please find our responses below.
>
> > Q1. Theorems 2 and 3 can benefit from a detailed discussion in the main body.
>
> A1. Theorem 2 justifies that when the model complexity and the sample size are large enough, the empirical solution given by our DAL risk will converge to its optimal value, i.e., $\min_\mathbf{w} R_D(\mathbf{w};\rho)$. Therefore, the difference between the expected and the empirical error is bounded w.r.t. the DAL risk.
>
> Theorem 3 goes one step further, studying the detection risk w.r.t. (unseen) real OOD data. It states that the open-world performance of our DAL depends on both the approximate risk and the estimation error. The former models the best performance (i.e., Bayes optimal) that our DAL can achieve, and the latter depends on the OOD distribution gap, the radius, and the excess error introduced in Theorem 2.
>
>
> In summary, **Theorem 2 considers the convergence for DAL itself, while Theorem 3 justifies that DAL can mitigate the OOD distribution discrepancy in the open world**. We will add the related discussion in the Appendix for our revision.
>
>
> > Q2. This could be a question as well - are common corruptions of CIFAR10 or Imagenet not used as standard datasets for OOD detection?
>
> A2. To our knowledge, ID feature corruptions are typically not considered in OOD detection [1]. Such anomalies may violate the standard definition of OOD data in that the ID features can still conceive ID semantics after common corruptions [2,5]. However, we are sure that detecting common corruption is also essential, which may motivate generalized OOD detection in the future that can handle varying abnormal cases. It remains an open question that we will study.
>
> > Q3. Line 88: Does this mean that the labels in ID and OOD are mutually exclusive? Could you please explain if this is a significant assumption.
>
> A3. Yes, the label space between ID and OOD data should be disjoint---a standard definition that clarifies the problem setup in OOD detection [2,5]. Such a definition is introduced in our paper for concreteness and clarity.
>
> > Q4. Line 124: Please provide references for the statement "the auxiliary and the real OOD data differ largely in practice". I would like to understand how significant is the typical difference.
>
> A4. It is a standard learning setup in OOD detection since the seminal work of outlier exposure [1] states, "In real applications, it may be difficult to know the distribution of outliers one will encounter in advance." It is well-adopted in the following works [3,4,5]. However, seldom work attempted to address the OOD distribution discrepancy issue in a systematic way, highlighting the contribution of our paper.
>
>
> > Q5. As a follow-up, how large can the upper bound in (4) grow? Are there any standard assumptions made for theoretical analysis?
>
> A5. For **sample sizes**, the generalization error decreases w.r.t. the reciprocal for the square root of sample sizes; For **distribution discrepancy**, the upper bound grows liearnly w.r.t. to the Wasserstein distance between auxiliary and real OOD data.
>
> Moreover, Eq. 4 is a particular case of Theorem 2 given $\rho=0$, thus following the same assumptions in Theorem 2, all commonly used in the literature [7]. In A6, we further explain that these assumptions are reasonable in practice.
>
> > Q6. What does the assumption in Theorem 2 regarding l <= M imply- suppose l is cross-entropy or exponential loss, does the bound mean that f_w needs to be a reasonably good solution? Otherwise, the loss function could be unbounded, if I am not mistaken. How mild are these assumptions?
>
> A6. One can prove that the cross-entropy and the exponential losses are bounded and lipschitz w.r.t. $\mathbf{w}$ for deep models with softmax outputs [6], if
>
> - (1) activation functions are 1-Lipschitz;
> - (2) inputs are from bounded feature space $\mathcal{X}$;
> - (3) parameter space $\mathcal{W}$ is bounded (e.g., using regularization to ensure).
>
> More specifically, when the F-norm bounds parameters, softmax output $\mathbf{f}$ is continuous and never attains infinity. If we further assume that inputs are from a bounded feature space, then $\mathbf{f}$ is a continuous function over the bounded space, implying that $\mathbf{f}$ has upper and lower bounds. **Thus the cross-entropy and the exponential loss can be bounded in practice and our assumptions are practical.**
>
> > Q7. It would be useful if the authors discussed a few most relevant open challenges, in light of their findings.
>
> A7. OOD distribution discrepancy is critical in OOD detection, yet little work has tried to handle it. Although DAL provides a systematic method to mitigate such a critical problem, the trade-off between the estimation and the approximation errors may limit further improvement. Moreover, other data augmentation approaches, which may lead to semantic diversity, are also open challenges. Our future studies will focus on novel learning schemes that can address the above issues.
>
> ---------------
>
> [1] Dan Hendrycks, Mantas Mazeika, and Thomas Dietterich. "Deep anomaly detection with outlier exposure." ICLR, 2018.
>
> [2] Zhen Fang, Yixuan Li, Jie Lu, Jiahua Dong, Bo Han, Feng Liu. "Is out-of-distribution detection learnable?" NeurIPS, 2022.
>
> [3] Yiyou Sun, Yifei Ming, Xiaojin Zhu, and Yixuan Li. "Out-of-distribution detection with deep nearest neighbors." ICML, 2022.
>
> [4] Yiyou Sun, Chuan Guo, and Yixuan Li. "React: Out-of-distribution detection with rectified activations." NeurIPS, 2021.
>
> [5] Yifei Ming, Ying Fan, and Yixuan Li. "POEM: Out-of-distribution detection with posterior sampling." ICML, 2022.
>
> [6] Noah Golowich, Alexander Rakhlin, and Ohad Shamir. "Size-independent sample complexity of neural networks." COLT, 2018.
>
> [7] Tong Zhang. "Mathematical analysis of machine learning algorithms."" Cambridge University Press, 2023.

---

> > ### Comment · Reviewer_REz1 · 2023-08-17
> > **Thank you for the response, increasing score**
> >
> > The authors have addressed my queries, thus I am glad to increase the score to 7. Thank you for a clear response.

---

### Comment · Area_Chair_oc5i · 2023-08-18

Dear Authors,

Thank you for your thorough and detailed responses to the reviewers' comments. We appreciate the efforts you have made to address each point raised by the reviewers and provide insightful explanations.

Your responses have been carefully reviewed, and we will take them into full consideration during our discussions.

Best regards,

Area Chair

---

### Decision · Program_Chairs · 2023-09-21

**Decision:**

Accept (poster)

**Comment:**

The paper presents DAL, a learning-based solution for out-of-distribution (OOD) detection in supervised learning. By expanding the set of auxiliary distributions and reducing distribution discrepancy, DAL outperforms existing techniques, as confirmed by positive feedback from all reviewers. The proposed method is well-motivated, has theoretical grounding, and demonstrates its effectiveness through extensive experiments. Given the resolution of raised concerns and the unanimous positive reviews, the paper is accepted for publication.